# Learning Structure from the Ground up— Hierarchical Representation Learning by Chunking

## Abstract

From learning to play the piano to speaking a new language, reusing and recombining previously acquired representations enables us to master complex skills and easily adapt to new environments. Inspired by the Gestalt principle of *grouping by proximity* and theories of chunking in cognitive science, we propose a hierarchical chunking model (HCM). HCM learns representations from non-i.i.d sequential data from the ground up by first discovering the minimal atomic sequential units as chunks. As learning progresses, a hierarchy of chunk representation is acquired by chunking previously learned representations into more complex representations guided by sequential dependence. We provide learning guarantees on an idealized version of HCM, and demonstrate that HCM learns meaningful and interpretable representations in visual, temporal, visual-temporal domains and language data. Furthermore, the interpretability of the learned chunks enables flexible transfer between environments that share partial representational structure. Taken together, our results show how cognitive science in general and theories of chunking in particular could inform novel and more interpretable approaches to representation learning.

## 1    Introduction

The last decade has witnessed a meteoric rise in the abilities of artificial systems, in particular deep learning models (LeCun et al., 2015). From beating the world champion of Go (Silver et al., 2016) to predicting the structure of the human proteome (Jumper et al., 2021), deep learning models have accomplished impressive end-to-end learning achievements. Yet, researchers were quick to point out shortcomings (Marcus, 2018; Lake et al., 2017). Two such shortcomings concern the hierarchical structure and interpretability of the representations learned by neural network architectures. Specifically, deep learning models suffer from a lack of interpretability since ANNs are sub-symbolic, nested, non-linear structures. This means they the way predictions are generated can be difficult to understand (Samek et al., 2017; Ribeiro et al., 2016; Doshi-Velez & Kim, 2017). Furthermore, deep learning models can also struggle to learn hierarchical representations altogether (Lake & Baroni, 2018; Fodor & Pylyshyn, 1988). To address these shortcomings, it has been suggested that machine learning researchers could seek inspiration from cognitive science and construct models that resemble the hierarchical and interpretable representations observed in human learners (Chollet, 2017; Lake et al., 2015).

We take inspiration from the cognitive phenomenon of *chunking* and the Gestalt principle of *grouping by proximity*. To get an intuition for chunking, try to read through the following sequence of letters: "DFJKJKJKDFDFJKJKDFDF". Upon reaching the end, if you were tasked to repeat the letters from memory, you might recall fragments of the sequence such as "DF" or "JK". By parsing the sequence of letters only once, one detects frequently occurring patterns within and memorize of them together as units, i.e. *chunks*. Chunking has been observed in a range of sensory modalities. We perceive units and structures when learning a language (Perruchet et al., 2014; McCauley & Christiansen, 2017), in action sequences (Penhune & Steele, 2012; Rosenbaum et al., 1983), and in visual structures Hinton (1979); Brady et al. (2009); Egan & Schwartz (1979). The extracted chunks have been argued to facilitate memory compression (Gobet et al., 2001; Miller, 1956), compositional

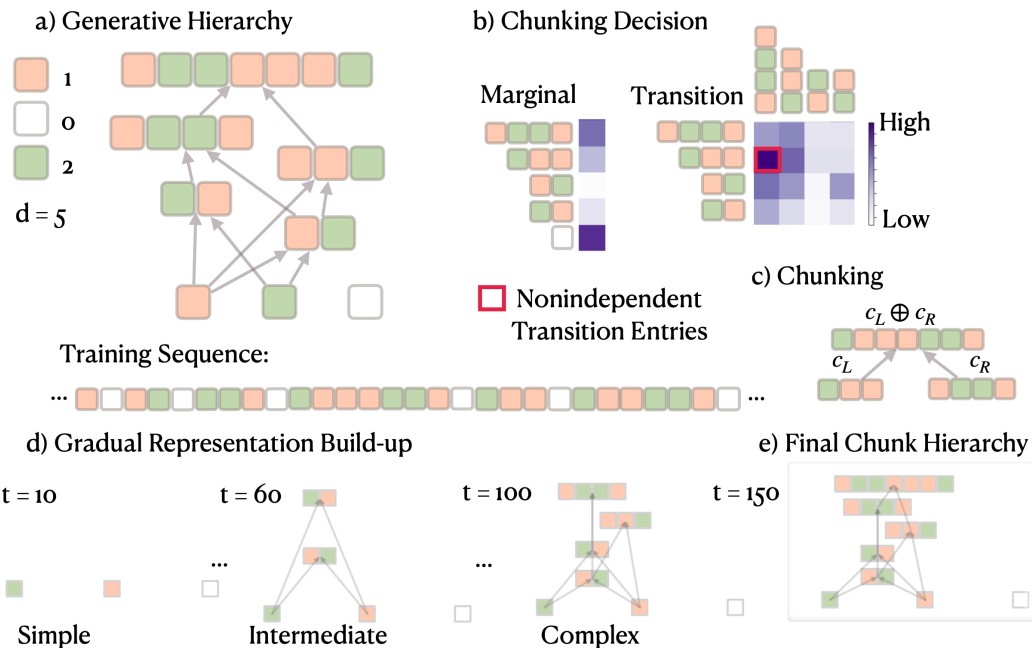

Figure 1: Schematic of the Hierarchical Chunking Model. **a)** Example of a hierarchical model generating training sequences. **b)** Intermediate representation of learned marginal and transition matrices. The most frequent transition that violates the independence testing criterion is marked in red and can be turned into a new chunk. **c)** HCM combines the two chunks $c_L$ and $c_R$ to form a new chunk. **d)** As HCM observes longer sequences, it gradually learns a hierarchical representation of chunks. **e)** HCM arrives at the finally chunk hierarchy isomorphic to the generative hierarchy.

generalization (Schulz et al., 2017), predictive processing (Koch & Hoffmann, 2000; Müssgens & Ullén, 2015), and communication (Schulz et al., 2020).

In the current work, we propose a hierarchical chunking model (HCM) that learns chunks from non-i.i.d sequential data with a hierarchical structure. HCM starts out learning an atomic set of chunks to explain the sequence and gradually combines them into increasingly larger and more complex chunks, thereby learning interpretable hierarchical structures. The output of the model is a dynamical graph that is a trace of the evolving representation. The resulting representations are therefore easy to interpret, and flexibly reusable (e.g. we can choose to re-use specific parts). We derive learning guarantees on an idealized generative model and demonstrate convergence on sequential data coming from this generative model. Furthermore, we show that our model can benefit from having learned previous structures with shared components, leading to flexible transfer and generalization. Finally, we demonstrate that HCM can return interpretable representation in discrete sequential, visual, and language domains, in several small-scale experiments. Taken together, our results show how cognitive science in general and theories of chunking and grouping by proximity in particular can inform novel and more interpretable approaches to representation learning.

## 2    HIERARCHICAL CHUNKING MODEL

When talking about chunks in a sequence, a chunk is defined as a unit created by concatenating several atomic sequential units together. Take the training sequence shown in Figure 1a as an example. The sequence consists of discrete, size-one atomic units from an atomic alphabet $\mathbb{A}_0$: in this case $\mathbb{A}_0 = \{0, 1, 2\}$. A chunk $c$ is made up of a combination of one or more atomic units in $\mathbb{A}_0 \backslash \{0\}$. 0 denotes an empty observation in the sequence.

Intuitively, if the training sequence contains inherent hierarchical structure, then there are patterns which span several sequential units sharing these internal structures, like repeated melodies and sub-melodies in a piece of music. If the pattern occurs in the sequence, observations between sequential units within the pattern will be correlated. In this case, chunking patterns within a sequence as units simplifies perceptual processing in the sense that the sequence can be perceived one chunk at

a time, instead of one sequential unit at a time. Furthermore, the acquired "primary" chunks serve as building blocks to discover larger chunks that are embedded within the intricate hierarchy of the sequential structure.

More formally, HCM acquires a belief set $\mathbb{B}$ of chunks, and uses chunks from the belief set to parse the sequence one chunk at a time. HCM assumes that a sequence is generated from samples of independently occurring chunks with probability of $P_{\mathbb{B}}(c)$ evaluated on the belief set $\mathbb{B}$. The probability of observing a sequence of parsed chunks $c_1, c_2, ..., c_N$ can be denoted as $P(c_1, c_2, ..., c_N) = \prod_{c_i \in \mathbb{B}} P_{\mathbb{B}}(c_i)$. Chunks as perceiving units serve as independent factors that disentangle observations in the sequence.

From the beginning of the sequence, HCM 'perceives' sequential units one chunk at a time, in other words, the training sequence is parsed by HCM into chunks. At every parsing step, the longest chunk in the belief set that is consistent with the upcoming sequence is chosen to explain the up-coming sequential observations. The end of the previous chunk parse initiates the next parse.

Observing a hierarchically structured sequence as illustrated in Figure 1a, HCM gradually builds up a hierarchy of chunks starting from an empty belief set. It first identifies a set of atomic chunks to construct its initial belief set $\mathbb{B}$. Initially, these will be chunks of length one, yielding one-by-one processing of the primitive elements.

For one belief set $\mathbb{B}$, HCM keeps track of the marginal parsing frequency $\boldsymbol{M}(c_i)$ for each chunk $c_i$ in $\mathbb{B}$, a vector with size $|\mathbb{B}|$ and the transition frequency $\boldsymbol{T}$ between chunk $c_i$ followed by chunk $c_j$, as illustrated in Figure 1b. Entries in $\boldsymbol{M}$ and $\boldsymbol{T}$ are used to test the hypothesis that consecutive chunk parses have a correlated consecutive occurrence within the sequence. If two chunks $c_L$ and $c_R$ have a significant adjacency dependence based on their entries in $\boldsymbol{M}$ and $\boldsymbol{T}$, they are chunked together to become $c_L \oplus c_R$, which augments the belief set $\mathbb{B}$ by one. One example of chunk merging is shown in Figure 1c.

There are two different versions of HCM. The Rational Chunk Learning HCM produces chunks in an idealized way which we use to study learning guarantees. The online version of HCM is an approximation to the rational HCM that can be adapted to different environments and processes sequences online. Pseudo-code for both algorithms can be found in Algorithm 1 and 2 in the SI.

**Rational Chunk Learning: HCM as an Ideal Observer** The model is initiated with an empty belief set, and it first finds a minimally complete belief set after the first sequence parse. In each iteration, the entire sequence is parsed to evaluate $\boldsymbol{M}$ and $\boldsymbol{T}$, which are used to find consecutive chunk parses in the existing belief set fulfilling the dependence testing criterion. From these dependent chunk pairs, the pair with the largest estimated joint probability is combined into a new chunk. The new chunk enlarges the belief set by one. The chunks in the new belief set are used to parse the sequence in the next iteration. This process repeats until all of the chunks in the belief set pass an independence testing criterion.

**Online Chunk Learning** The online chunk learning HCM approximates the ideal observer HCM by learning new chunks when the training sequence is processed online. To have a feature that encourages adaptation to new environmental statistics, entries in $\boldsymbol{M}$ and $\boldsymbol{T}$ can be subject to memory decay. We use the ideal observer model to demonstrate learning guarantees, but use the online model to learn representations in realistic and more complex set-ups.

## 2.1 HCM BUILDS REPRESENTATIONS FROM THE GROUND UP

As HCM learns from a sequence, it starts with no representation and gradually builds up the representations which can be readily interpreted at any learning stage. The representation can be described by a chunk hierarchy graph $\hat{\mathcal{G}}$ with the vertex set being the chunks and edges pointing from chunk constituents to the constructed chunks. Shown in Figure 1d is the gradual build-up of one such chunk hierarchy graph as the model learns from a training sequence coming from the generative hierarchy displayed in Figure 1a. At $t = 10$, HCM learns only the atomic chunks, at $t = 60$, HCM has already constructed two additional chunks; when $t = 100$, two more additional chunks are constructed based on the previously acquired chunks, and at $t = 150$, HCM arrives at the final chunk hierarchy.

Figure 2: **a)** Example graph generated from the hierarchical generative model with a depth of $d = 3$. **b)** Learning performance of HCM and RNN with increasing training length and for different depths. Performance was averaged over 30 randomly-generated graphs.

# 3 HIERARCHICAL GENERATIVE MODEL

To study learning guarantee of the HCM, a generative model with a random hierarchy of chunks is constructed. The relation between chunks and their constructive components in the generative model is described by a chunk hierarchy graph $\mathcal{G}_d$ with vertex set $V_{\mathbb{A}_d}$ and edge set $E_{\mathbb{A}_d}$. One example is illustrated in Figure 1a. $\mathbb{A}_d$ is the set of chunks used to construct the sequence. The depth $d$ specifies the number of chunks created in the generative process.

Starting with an initial set of atomic chunks $\mathbb{A}_0$, at the i-th iteration, two chunks $c_L, c_R$ are randomly chosen from the current set of chunks $\mathbb{A}_i$ and are concatenated into a new chunk $c_L \oplus c_R$, augmenting $\mathbb{A}_i$ by one to $\mathbb{A}_{i+1}$. Meanwhile, an independent occurrence probability is assigned to each chunk under the condition that the probability of occurrence for every new chunk $c_i$ in the construction process evaluated on the support set $\mathbb{A}_i$ carries the largest probability mass.

To generate a sequence from a constructed hierarchical graph, chunks are sampled independently from the set of all chunks and appended after each other, under the constraint that no two chunks with a child chunk should be sampled consecutively.

## 3.1 LEARNING GUARANTEE

**Theorem**: As the length of the sequence approaches infinity, HCM learns a hierarchical chunking graph $\hat{\mathcal{G}}$ isomorphic to the generative hierarchical graph $\mathcal{G}$.

*Proof Sketch*: We approach this proof by induction. Further details can be found in SI. Base step: The first step of the rational chunking algorithm is to find the minimally complete atomic set of chunks to form its initial belief set. This procedure guarantees that $\hat{\mathcal{G}}_0 = \mathcal{G}_0$. Additionally, the probability mass of the learning model at step 0 and the generative model at step 0 is asymptotically the same as the sequence length approaches infinity. Induction hypothesis: Assume that the learned belief set $\mathbb{B}_i$ at step $i$ contains the same chunks as the alphabet set $\mathbb{A}_i$ in the generative model, the chunk combination pair with the biggest evaluated joint occurrence probability violating the independence test is picked to be concatenated into a chunk to extend the belief set: this chunk is the same chunk node created by the hierarchical generative model. End step: The chunk learning process stops once the independence criterion is passed. This is the case once the chunk learning algorithm has learned a belief set $\mathbb{B}_d = \mathbb{A}_d$.

## 3.2 LEARNING CONVERGENCE

HCM's learning performance with increasing sequence length is evaluated and shown in Figure 2. For this, HCM was trained on random graph hierarchies constructed from the hierarchical generative model over 3000 trials while also varying the depth $d$. Figure 2a displays an example of a random generative hierarchy with a depth of $d = 3$. We used the Kullback-Leibler divergence to evaluate the deviation of HCM's learned representations from the generative hierarchical model. This was done by using HCM's representation to produce "imagined" sequences, which were then compared to the ground truth probability for each chunk in the used generative model. Figure 2b shows the KL-divergence between the learned and ground-truth distribution for different depths $d$ of the generative graphs. For each depth, the KL-divergence is evaluated on 30 random generative models with sequence length increasing from 50 to 3000 in each model. Overall, the KL-divergence decreased as the length of training sequence increased and converged with larger training sequences. This shows empirically that HCM learns representations bearing closer resemblance to the generative model.

Figure 3: **a)** Example of a representation learned by an HCM. **b)** An environment facilitative to the learned representation. Gray shadows mark the chunks that can be directly transferred. **c)** Average performance over the first 500 trials after environment change in the facilitative environment. **d)** Interfering environment. Gray shadows marks chunks that the learned representation needs to acquire from scratch. **e)** Average performance over the first 500 trials after environment changes into an the interfering environment.

We used the same sequences to train a 3-layer Recurrent Neural Network (RNN) with 40 hidden units and the same method to measure the KL-divergence by using "imagined" sequences produced by the RNN evaluated on the support set of the generative hierarchy. As the length of the training sequence increased, the KL-divergence also converged but at a much slower rate. Note that HCM's competitive advantage increased further as the depth of the generative hierarchy increased. Thus, HCM learns quickly about the hierarchical generative model and does so faster than a neural network.

### 3.3 HCM PERMITS TRANSFER BETWEEN ENVIRONMENTS WITH OVERLAPPING STRUCTURE

After training on a sequence, HCM acquires an interpretable representation. Knowing what the model has learned enables us to directly know what type of hierarchical environment would facilitate or interfere with the learned representations. This is impossible to do using standard neural networks because we generally would not know what representation they have learned, and therefore do not have fine-grained control over retaining or removing parts of it.

More formally, two HCM models might have acquired different hierarchical chunking graphs $\mathcal{G}_i$ and $\mathcal{G}_j$ from their past experience. These might lie on the graph construction path $(\mathcal{G}_0, \mathcal{G}_1, \mathcal{G}_2, ..., \mathcal{G}_d)$. The HCM with a chunk hierarchy graph 'closer' to the ground truth chunk $\mathcal{G}_d$ on the path, takes fewer iteration to arrive at $\mathcal{G}_d$. This also applies when the chunk hierarchies starting out are not along the graph construction path but only showing partial overlap.

Similarly, the chunk hierarchy $\mathcal{G}_i$ learned by an HCM might facilitate its performance in a new environment where $\mathcal{G}_i$ lies along the graph construction path to the true $\mathcal{G}_d$, i.e. there is partial overlap between the chunk hierarchies. We demonstrate this in Figure 3. Here, HCM was trained on a random hierarchical generative model and acquired the representation shown in Figure 3a. Knowing this representation, we also know that an environment with a hierarchical generative graph as shown in Figure 3b allows for positive transfer, i.e. the HCM that already contains the chunk hierarchy Figure 3a would learn faster than a naive one by transferring its previously learned chunks. As shown in Figure 3c, the trained HCM learns faster than a naive HCM which had to learn about the structure from scratch.

Vice versa, we might have a situation in which transfer is detrimental. For example, there is no overlap in the chunk hierarchy learned by the HCM in Figure 3a with the graph shown in Figure 3d. The shaded chunks need to be learned anew by the HCM trained on Figure 3a, yet the previously acquired chunks might mislead HCM causing it to adapt to the new environment more slowly. As a result, the performance of the pre-trained HCM suffers more from an interfering environment than a naive HCM (Figure 3e). This is similar to catastrophic interferences in neural networks (Sharkey & Sharkey, 1995), where performance on one task interferes with others. The advantage of the HCM is that with visibility into the chunks learned, we have a better a priori sense of whether transfer will be facilitative or interfering. This means that we can examine whether to re-use the previously learned chunks, or to start from scratch with a naive model.

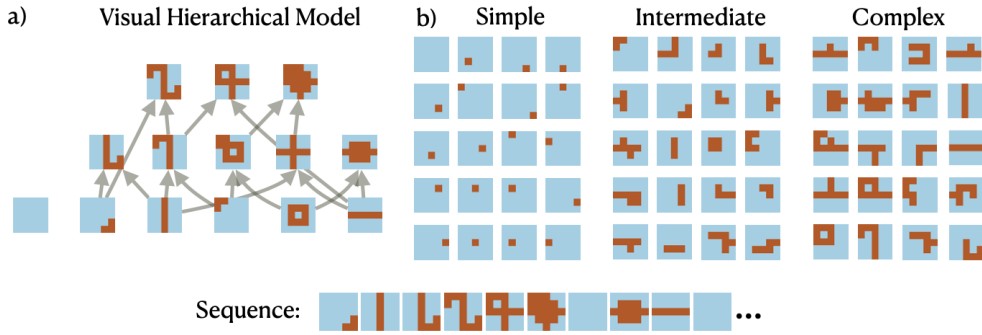

Figure 4: **a)** A designed visual hierarchical model where elementary components compose more complex images. **b)** Initial, intermediate, and complex chunks learned by HCM trained on sequences of images generated by the visual hierarchical model.

# 4    GENERALIZING TO VISUAL TEMPORAL CHUNKS VIA THE PRINCIPLE OF PROXIMAL GROUPING

Humans are not only able to identify sequential chunks but also to find structure in visual-temporal stimuli and group visual points as a whole. For example, despite the fact that our retina receives pixel-wise visual inputs that vary across temporal slices, we are able to perceive complex moving objects. The Gestalt principle of *grouping by proximity* states that objects that are close to one another appear to form groups (Wagemans et al., 2012). This principle has been argued to play a key role in human perceptual grouping (Compton & Logan, 1993) and benefit working memory (Peterson & Berryhill, 2013) and the reduction of visual complexity (Donderi, 2006). Indeed, in humans and other animals, learning of adjacent relationships prevails over non-adjacent ones (Malassis et al., 2018). Therefore, the adjacent dependency structure can be seen as the primary driver of chunking in visual temporal domains (Froyen et al., 2015). To emulate this ability of chunking via proximal grouping in visual temporal perception, we extend HCM to also learn visual temporal chunks.

Visual temporal chunks not only subsume temporal length but also varying visual slices in each temporal slice. One can imagine visual temporal chunks as having a 3D shape — the first two height and width dimensions are the visual part of the chunk, the length of the object is the temporal part, made of stacked visual-temporal pixels. Within each temporal slice are the visual features identified by the chunk. Since nearby points are more likely to be in the same chunk, this assumption is an implementation of the principle of grouping by proximity in the visual-temporal domain. As the model iterates through data across its temporal slice, the chunk that attains the biggest visual temporal volume is chosen to explain parts of the visual-temporal observations. Multiple visual temporal chunks can be identified to occur simultaneously. Starting at the visual temporal time point marked by the previous chunk, chunks are identified and stored in $M$. The transition matrix $T$ is modified to account for the temporal lag-difference between adjacent chunk pairs and records the frequency that one chunk transitions into another one with a given time-lag. A hypothesis test is conducted every time when a pair of adjacent chunks are identified, chunks that violate the hypothesis test are grouped together to augment the belief set which is then used to parse future sequences.

## 4.1    LEARNING PART-WHOLE RELATIONSHIP BETWEEN VISUAL COMPONENTS

We let HCM learn chunks in a visual domain by learning from a sequence of images. To this end, a hierarchical generative model in the pixel-wise image domain shown in Figure 4a was constructed to test HCM's visual chunking ability. Specifically, a set of elementary visual units in the lowest level of the hierarchy are combined to construct intermediate and more complex visual units higher up in the hierarchy. An empty image is included to denote pauses. All of the constructed elements in the hierarchy occurred independently according to a draw from a Dirichlet flat distribution. To generate the sequence which was used to train HCM, images in the hierarchy were sampled from the generative distribution and appended to the end of the sequence. In this way, despite the visual correlations in each image described by the hierarchy, each image slice was temporally independent from other slices.

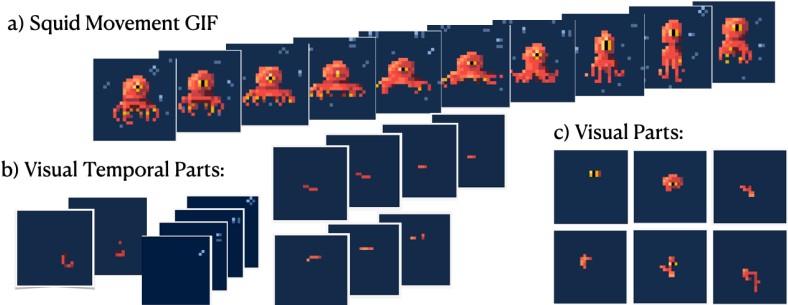

Figure 5: **a)** A GIF of a moving squid used as a sequence to train HCM. **b)** Examples of temporal chunks learned by HCM. **c)** Examples of visual chunks learned by HCM.

HCM learns a hierarchy of visual chunks from a sequence of visually correlated but temporally independent images sampled from the visual hierarchical model. Shown in Figure 4b are the chunk representations learned by HCM at different stages. Having no knowledge about the image parts before starting to learn, HCM acquires the individual pixels as chunks to explain the observations. As HCM proceeds with learning, visual correlations among the pixels are discovered and larger chunks are formed. As the number of observations increases, the model learns more sophisticated yet still interpretable chunks.

## 4.2 LEARNING VISUAL-TEMPORAL MOVEMENT HIERARCHIES

Instead of seeing one image after another sampled from an independent, identically distributed distribution, real world experiences contain correlations in both the visual and temporal dimension. From observing object movements across space and time, the visual system learns structures from correlated visual and temporal observations, decomposes motion structure and groups moving objects together as a whole (Bill et al., 2020). To emulate this type of environment as a learning task, an animated GIF of a squid swimming in the sea as shown in Figure 5a was used as a visual-temporal sequence to train HCM. As learning advances, HCM learns chunks spanning both the visual and temporal domain. Examples of such visual-temporal chunks are shown in Figure 5b and c. There are visual-temporal chunks that mark movements of a tentacle and the rising-up motion of a bubble. Additionally, there are visual chunks that resemble a part of the squid's eye and face.

## 5 LEARNING CHUNKS FROM REALISTIC LANGUAGE DATA

We so far have only shown demonstrations of HCM learning chunks on simple sequences. Thus, one step further is to run HCM on real world datasets that contain complex hierarchical structures. One immediate testbed containing such structures is natural language. To this end, we trained HCM to learn chunks on the first book of *The Hunger Games*.

HCM starts with learning individual English letters and punctuation. After having seen 10,000 characters of the book, HCM acquires frequently used chunks that resemble common English pre- and suffixes such as "ity", "ing" , "re", and "ith"; definite and indefinite articles such as "a", "an", "the"; conjunctions such as "and", "but", "that", and "as"; prepositions such as "of", "to", "in", "at", variants of "is", "are", "was"; and pronouns such as "he", "my", "me", "we", "she".

After having seen 100,000 characters of the book, HCM learns intermediate chunks that include various commonly used verbs such as "come", "sing", "leave", "try"; nouns such as "bed", "day", "mother"; common word combinations such as "in the", "lose to", "the wood". Additionally, HCM has acquired words specific to *The Hunger Games* such as "prim", "death", "hunger", and "district 12".

After having seen 300,000 characters of the book, HCM learns more complex phrases from the previously learned words, such as the commonly used phrases "it is not just", "in the school", "our district", and "cause of the".

Table 1: Example Chunks Learned from *The Hunger Games*

| Simple Chunks | 'an', 'in ', 'be', 'at', 'me', 'le', 'a ', 'ar', 're', 'and ', 've', 'ing ', 'on', 'st', 'se', 'to ', 'of ', 'he', 'my ', 'te', 'pe', 'ou', 'we', 'ad', 'de', 'li', 'the', 'ce', 'is', 'as', 'il', 'ch', 'al', 'no', 'she', 'ing', 'am', 'ack', 'we', 'raw', 'on the', 'day', 'ear', 'oug', 'bea', 'tree', 'sin', 'that', 'log', 'ters', 'wood', 'now', 'was', 'even', 'leven', 'ater', 'ever', 'but', 'ith', 'ity' |
|---|---|
| Intermediate Chunks | 'this', 'pas', 'eak', 'if', 'sing', 'bed', 'men', 'raw', 'day', 'in the', 'link', 'for', 'one', 'the wood', 'bell', 'other', '...', 'lose to', 'hunger', 'mother', 'death', 'would', 'district 12', 'try', 'under', 'prim', 'beg', 'then', 'into', 'gale', 'read', 'come', 'he want', 'leave', 'where', 'older', 'says', 'might', 'dont', 'add', 'know', 'man who', 'of the' |
| Complex Chunks | 'out of', 'it out', 'our district', 'capitol', 'reaping', 'fair', 'berries', 'the last', 'fish', 'again', 'as well', 'the square', 'scomers', 'fully', ', but the ', 'in the school', 'at the', 'you can', 'tribute', 'to remember', 'it is not just', 'I can', 'peace', 'feel', 'you have to', 'I know', 'bother', 'in our', 'kill', 'cause of the', 'the pig', 'to the baker', 'I have', 'what was' |

## 6 RELATED WORK

Our model is successor to decades of work on different approaches to chunk learning. In cognitive science, researchers have put forward models that produce qualitatively similar chunks as humans learning a natural or artificial languages (Servan-Schreiber & Anderson, 1990; Perruchet & Vinter, 1998), as well as models that can extract chunks from visual inputs (Mareschal & French, 2016). HCM can be viewed as a principled version of these earlier cognitive models because it uses hypothesis testing in its decision to chunk elements together instead of using chunking heuristics. Therefore, HCM comes with learning guarantees for a fairly general class of hierarchically structured environments. Additionally, HCM extends past cognitive models to the higher dimensional domain of visual-temporal chunking.

The primary approach to chunk learning in the language domain were $n$-gram models, dating all the way back to Shannon (1948). An $n$-gram model learns the marginal probabilities of all chunks of size $n$. A major limitation of this approach is that the number of chunks grows exponentially as a function of chunk length. For instance, with an alphabet of 26 letters, the number of possible 5-letter words already goes beyond 10 million. Due to the large vocabulary size, building word-level $n$-gram models is virtually unfeasible. To this end, a Bayesian non-parametric extension of $n$-gram models has been put forward (Teh, 2006). A Bayesian non-parametric $n$-gram model builds up structure as evidence is accumulated. That is, it flexibly reduces to a 1-gram model if chunks are not present or 'opens up' higher and higher $n$-gram levels depending on the size of chunks. Teh (2006)'s model is different from HCM in several regards. Instead of using the chunks to 'look forward' and parse the sequence in large steps, it 'looks back' and predicts only one element conditioning on the context of the previous elements. Then, instead of storing an explicit bag of chunks, it represents a chunk distribution weighted by evidence. For prediction, it smooths over the evidence of all chunks that are consistent with the current context. A shortcoming of Teh (2006)'s model and $n$-gram models in general is that they do not leverage the concatenation process observed in humans but the chunks are built up primitive element-wise.

Hierarchical hidden Markov models (Fine et al., 1998) were developed in a similar vein, extending hidden Markov models to be able to capture multi-scale sequential structure. Although these models are able to capture larger patterns than non-hierarchical versions while maintaining the computational tractability of simple Markov processes, they still lack the adaptive recombination and reuse of pre-existing components. Fragment grammars (O'Donnell et al., 2009) address this by balancing the creation and re-use of chunks by Bayesian principles, while also preserving the symbolic interpretability. However, fragment grammars are intractable and their inference is exceptionally costly.

In the era of deep learning, both natural language processing and image processing became increasingly dominated by neural networks, with one of their primary tasks being the extraction and prediction of chunks (Zhai et al., 2017; Si et al., 2020; Ortmann, 2021). Commonly, these so-called

sequence chunks are used as units of segmentation for other downstream tasks such as text translation. However, in these approaches, the architecture needs to be pre-specified before training as compared to HCM, where no such specification is required. HCM can therefore be seen as an interpretable and transparent alternative to neural networks for some of these tasks.

A final related line of research attempts to learn explicit representations by inducing programs (Lake et al., 2015). Fore example, in a recent approach to this challenge, the interpretable structures returned by program induction algorithms was combined with a deep neural network to learn meaningful representations from data (Ellis et al., 2021). Yet in these approaches the retrieved representations are highly dependent on the initial set of building blocks which are usually specified by the experimenter. HCM, in comparsion, is task-agnostic and requires no primitive program description.

# 7 DISCUSSION

We have proposed the Hierarchical Chunking Model (HCM) as a cognitively-inspired method for learning representations from the ground up. HCM starts out by learning atomic units from sequential data which it then chunks together, gradually building up a hierarchical representation that can be expressed as a dynamical and intepretable graph. We have provided learning guarantees for an idealized version of HCM, shown how HCM's interpretability can facilitate generalization, and demonstrated that HCM learns meaningful representations from visual, temporal, visual-temporal, and language data.

Although we have showcased HCM's abilities across a set of diverse experiments, some challenges remain. First of all, HCM is currently not computationally efficient, such that we needed to run the online version for most of the presented tasks. There are several directions that could enhance HCM's computational efficiency. One method to speed up learning could be to stitch multiple chunks together in one decision. Another direction could be to have the chunk decision process between all of the acquired chunks separated from the process of parsing observations, which could be used for parallelized implementations. We believe that scaling HCM up will be beneficial to learn in increasingly more complex data sets than the ones we have applied here.

Secondly, all patterns of chunks in the current project came from adjacent events in the sequences. This feature was inspired by the Gestalt principle of grouping by proximity. Yet many patterns observed in natural data sets might exhibit non-adjacent dependencies in space or time. The adjacency assumption therefore limits the model from detecting such patterns. How to relax the adjacency assumption as a grouping criterion to allow for non-adjacent relationships to be chunked together remains an open challenge. Furthermore, HCM currently assumes that there is a hierarchy of chunks which occur independently in the observational sequence. This set-up was intended to be a simplifying assumption for a first approach toward building a cognitively plausible hierarchical chunking model. Nonetheless, our approach can and should be extended to more sophisticated assumptions such as higher order Markovian dependencies between chunks. Finally, HCM passes visual data "as is" and does not take into consideration any additional assumptions about visual inputs such as translation or rotational symmetries, which humans can detect when perceiving visual structures.

There are also several avenues for future investigations. One immediate step is to run HCM on other, more complex data sets such as musical scores, neural data, and large natural language corpora, to name but a few. Furthermore, we intend to not only run HCM on raw visual inputs directly but also to employ neural networks to compress inputs into a latent space and then train HCM on these latent dimensions (Franklin et al., 2020). Additionally, one could also use deep learning models to label parts of objects and then train HCM on the movements of these labelled parts (Insafutdinov et al., 2016). Finally, we are currently only testing for independence when deciding on whether or not to chunk, although other statistical tests are conceivable. One general class of tests could be to assess whether or not chunks are causally related with each other, in an attempt to find the best causal structure to explain the sequential, observational data Heinze-Deml et al. (2018). This would make HCM a useful model of causal representation learning (Schölkopf et al., 2021).

## 8 Reproducibility Statement

Detailed information about the HCM algorithm, proof, generative model, independence test and experimental details and results can be found in the supplementary information section. The code used for the algorithm and experiments will be available as a comment to the reviewers and area chairs as a link to an anonymous repository as soon as the discussion forum for all submitted papers is open.

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

## A  INDEPENDENCE TEST

We use hypothesis testing on the assumption of independence to decide whether there exists an association between consecutive parses of any two chunks $c_L$ and $c_R$ in the current belief set. If the independence test is violated, then the two chunks are combined together. Another independence test is used to evaluate if there are still statistical associations between chunk observations for each possible chunks in the current belief set; this is used as a criterion to continue or halt the chunking process.

We use a $\chi^2$-test of independence to assess if the consecutive parses of $c_L$ and $c_R$ observed in $\boldsymbol{T}$ violate the independence criterion. Observations of $c_L$ and $c_R$ in parses are categorical variables and can be represented as rows and columns of a contingency table. The number of observations that $c_L = 1$ or any other observations ($c_L = 0$) consists of the row entries, indicating observations of $c_L$, while the number of observations $c_R = 1$ and $c_R = 0$ make up the column entries. The table, therefore, consists of two rows and two columns.

The null hypothesis is that the occurrence of the consecutive observations is statistically independent. Given the hypothesis of independence, the theoretical frequency for observing $c_L$ followed by $c_R$ is $E[c_L = 1, c_R = 1] = Np(c_L = 1)p(c_R = 1)$, with $N$ being the total number of parses. $p(c_L = 1) = \frac{N(c_L)}{N} = \frac{N(c_L \to c_R)}{N} + \frac{N(c_L \to \neg c_R)}{N}$, $p(c_R = 1) = \frac{N(c_R)}{N} = \frac{N(c_L \to c_R)}{N} + \frac{N(\neg c_L \to c_R)}{N}$

$$
\begin{aligned}
\chi^2 &= \sum_{l=\{0,1\}} \sum_{r=\{0,1\}} \frac{N(c_L = l, c_R = r) - E[c_L = l, c_R = r]}{E[c_L = l, c_R = r]} \\
&= N \sum_{l=\{0,1\}} \sum_{r=\{0,1\}} p(c_L = l)p(c_R = r)\left(\frac{(\frac{N(c_L = l, c_R = r)}{N}) - p(c_L = l)p(c_R = r)}{p(c_L = l)p(c_R = r)}\right)
\end{aligned}
\tag{1}
$$

The degree of freedom for this test is 1. A $\chi^2$-probability of less than or equal to 0.05 is used as a criterion for rejecting the hypothesis of independence.

The independence test is also employed to evaluate whether there are still statistical associations between chunk observations for each possible chunk in the current belief set, which we use as a criterion to continue or to halt the chunking process. For this test, the contingency table contains rows and columns corresponding to all possible chunks in the current belief set, and the $\chi^2$-statistic is calculated as:

$$
\chi^2 = \sum_{c_L \in sB} \sum_{c_R = sB} \frac{N(c_L, c_R) - E[c_L, c_R]}{E[c_L, c_R]} = N \sum_{c_L \in sB} \sum_{c_R = sB} p(c_L)p(c_R)\left(\frac{\frac{N(c_L, c_R)}{N} - p(c_L)p(c_R)}{p(c_L)p(c_R)}\right)
\tag{2}
$$

The degrees of freedom are $(|sB| - 1) * (|sB| - 1)$, and a $p$-value of $p \leq 0.05$ is again used as a criterion to reject the null hypothesis and therefore as evidence to continue the chunking process.

## B    RATIONAL CHUNKING ALGORITHM

---

**Algorithm 1:** Rational Chunking Algorithm

---

**input** : Seq, maxIter
**output:** $\mathbb{B}_d, \hat{\mathcal{G}}_d, \boldsymbol{T}_d, \boldsymbol{M}_d$
$d \leftarrow 0, iter \leftarrow 0$;
$\mathbb{B}_d, \boldsymbol{M}_d, \boldsymbol{T}_d$ = getSingleElementSets(Seq);      /* minimally complete atomic set */
**while** *!IndependenceTest($\boldsymbol{M}_d$, $\boldsymbol{T}_d$) or iter $\leq$ maxiter* **do**
     $\boldsymbol{M}_d, \boldsymbol{T}_d$ = Parse(Seq, $\mathbb{B}_d$);
     $c_L, c_R \leftarrow None; MaxChunk, MaxChunkP \leftarrow None; PreCk = \{\}$;
     **for** $(c_i, c_j) \in \mathbb{B}_d \backslash \{0\} \times \mathbb{B}_d \backslash \{0\}$ **do**
         $P_d(c_i \oplus c_j) = CalculateJoint(\boldsymbol{M}_d, \boldsymbol{T}_d, c_i, c_j)$;
         $P_{d+1}(c_i \oplus c_j) = \frac{P_d(c_i \oplus c_j)}{1 - P_d(c_i \oplus c_j)}$;
         **if** $P_{d+1}(c_i \oplus c_j) \geq$ *MaxChunkP and* $c_i \oplus c_j \notin PreCk$ *and !IndependenceTest($c_i$, $c_j$)*
         **then**
             $c_L \leftarrow c_i, c_R \leftarrow c_j$;
             $MaxChunkP \leftarrow P_{d+1}(c_i \oplus c_j), MaxChunk \leftarrow c_i \oplus c_j$
         **end**
     **end**
     $c \leftarrow c_L \oplus c_R; \mathbb{B}_{d+1} \leftarrow \mathbb{B}_d \cup c; \hat{\mathcal{G}}_{d+1} \leftarrow$ AugmentGraph($\hat{\mathcal{G}}_d, (c_L, c), (c_R, c)$);
     $PreCk.add(c)$;
**end**

---

## C  Online HCM with Generalization to Visual-Temporal Sequences

HCM learns a chunk hierarchy graph $\hat{\mathcal{G}}$ by training on a visual-temporal sequence. The visual-temporal chunks can retain various continuous shapes and may not fill the entire space, which means that there are no jumps from any visual temporal pixel within a chunk to another. Within a chunk, there is always a path that connects any two visual temporal pixels.

The input chunk hierarchy graph $\hat{\mathcal{G}}$ could be an empty graph which corresponds to the case that the HCM model has not been trained yet, or it can be a chunk hierarchy graph that HCM has acquired from training. $M$ contains the frequency count of each chunk in the belief set $\mathbb{B}$. $T$ stores pairs of parsed adjacent chunks and their difference in temporal lag. The temporal lag is the time difference between the end of the previous chunk and start of the following chunk. The generative model of the world assumed by the chunk learning model is such that each visual temporal chunk occurs independently for each parse. It then keeps checking whether adjacent chunks in the visual and temporal domain violate the independence testing criterion. If so, then the visual/temporal adjacent chunks are grouped together. The constituent parts of a chunk remains in the belief set.

---

**Algorithm 2:** Visual-Temporal Chunking

---

**input** : Seq, $\hat{\mathcal{G}}, \theta, DT$
**output:** $\hat{\mathcal{G}}$
$M, T \leftarrow \hat{\mathcal{G}}.M, \hat{\mathcal{G}}.T$;
PreviousChunkBoundaryRecord $\leftarrow$ [];                    /* Record Chunk Endings */
ChunkTerminationTime.setall(-1);
**while** *Sequence not over* **do**
    CurrentChunks, ChunkTerminationTime =
     IdentifyTheLatestChunks(ChunkTerminationTime);
    ObservationToExplain $\leftarrow$ refactor(Seq, ChunkTerminationTime);
    **for** *Chunk in CurrentChunks* **do**
        **for** *CandidateAdjacentChunk in PreviousChunkBoundaryRecord* **do**
            **if** *CheckAdjacency(Chunk, CandidateAdjacentChunk)* **then**
                $M, T, \mathbb{B}, \hat{\mathcal{G}} \leftarrow$ LearnChunking(Chunk, CandidateAdjacentChunk.
                  $M, T, \mathbb{B}, \hat{\mathcal{G}}$);
            **end**
        **end**
        ChunkTerminationTime.update(CurrentChunks)
    **end**
    PreviousChunkBoundaryRecord.add(CurrentChunks);
    Forgetting($M, T, \mathbb{B}, \hat{\mathcal{G}}, \theta, DT$, PreviousChunkBoundaryRecord);
**end**

---

The pseudocode for the Visual-Temporal HCM is shown in Algorithm 2. The input can be a visual-temporal sequence, an i.i.d visual sequence, or a temporal sequence.

To process and update chunks online, HCM iterates through the visual temporal sequence, identifies chunks, and marks the termination time corresponding to each visual dimension stored in ChunkTerminationTime. ChunkTerminationTime is a matrix with its size being the same as the visual dimension that stores the end point in each visual dimension that the previous chunk finished. Because chunks could finish at different time points in each visual temporal dimension, the algorithm explain chunks starting at the end point in each visual temporal dimension when the last identified chunk is finished up until the current time point to identify current, on-going chunks. The chunks with the biggest visual temporal volume are prioritized to explain the relevant observations up to the current time point. As multiple visual temporal chunks can be identified to occur simoutaneously, CurrentChunks is a set that stores the identified chunk that has not reached its end point.

Once one or several chunks are identified to be ending at a time point, they are stored inside the PreviousChunkBoundaryRecord and their finishing time is updated for each visual pixel in ChunkTerminationTime. Corresponding entries in $M$ are updated upon once a chunk has ended. The chunks

that finishes after the start of the current chunk is checked with each current chunk on whether there is a visual temporal adjacency. Additionally, the independence test between adjacent chunk pairs are carried out for the decision to possibly merge chunks.

Once a pair of adjacent chunks $c_L$ $c_R$ violates the independence testing criterion, they are combined into one chunk $c_L \oplus c_R$. A new entry is created in $M$ with the estimated joint occurrence frequency for $c_L \oplus c_R$, which is subtracted from the marginal record of $c_L$ and $c_R$. The transition entries associated with $c_L \oplus c_R$ are initialized to be 1. Other chunks inherit the transition entries from $c_R$ to other chunks.

At each time step the visual temporal chunking algorithm does the following things:

1. Identifies the chunks biggest in volume that explain observation in each dimension from the time point when the last chunk ended to the current time point, mark their time point, and store them in the set of current chunks.

2. Identifies adjacent chunks in previous chunks with each of the currently ending chunk and updates their marginal and transition counts.

3. Modifies the set of chunks used to parse the sequence based on their adjacencies.

Entries in $M$ and $T$ are subject to memory decay at the rate of $\theta$. If any entry goes below the deletion threshold $DT$, their corresponding entries in $M$, $T$, $\mathbb{B}$ and $\hat{\mathcal{G}}$ are deleted.

# D  DETAILS TO THE PROOF OF RECOVERABILITY

## D.1  DEFINITIONS

An observational sequence is made up of discrete, integer valued, size-one elementary observational unit coming from an atomic alphabet set $\mathbb{A}_0$, where 0 represents the empty observation set.

One example of such an observational sequence $S$ is:

010021002112000...

The elementary observation units such as '0', '1', and '2' come from the atomic alphabet set $\mathbb{A}_0 = \{0, 1, 2\}$.

**Definition 1 (*Chunk*)**

*A chunk is composed of several non-empty, concatenated elementary observations, embedded in the sequence. That is, a chunk can be made up from any combination of elements in $\mathbb{A}_0 \setminus \{0\}$*

Examples of chunks from the observational sequence can be '1', '21', '211', '12', '2112', ... etc. 0 represents an empty observation in the sequence.

**Definition 2 (*Belief Set*)**

*A belief set is the set of chunks that the HCM uses to parse training sequences. The belief set is denoted as $\mathbb{B}$.*

An example of a belief set that that the model has learned from $S$ could be $\mathbb{B} = \{0, 1, 21, 211, 12, 2112\}$.

**Definition 3 (*Parsing*)**

*The parsing of chunks initiates from the beginning of the sequence. At every parsing step, the biggest chunk in the belief set that matches the upcoming sequence is chosen to explain the observation. The end of the previous chunk parse initiates the next parse.*

The sequence $S$ parsed by the model that uses the belief set $\{0, 1, 21, 211, 12, 2112\}$ results in the following partition. 0 1 0 0 21 0 0 2112 0 0 0.

We say that a belief set is *complete* if at any point when the model parses the sequence, the upcoming observations can be explained at least by one chunk within the belief set.

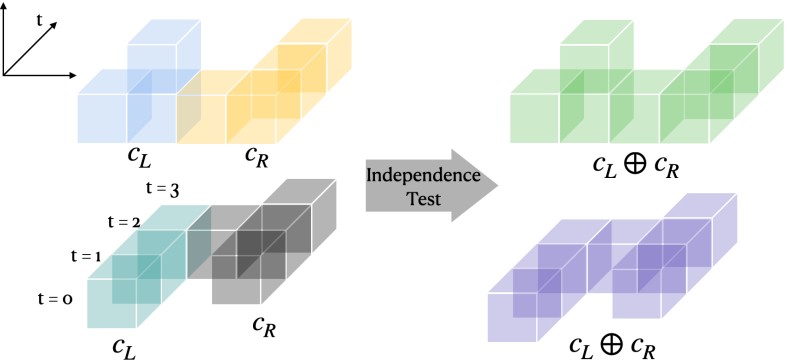

Figure 6: Illustration of Visual Temporal Chunks

In this work, we only refer to complete belief sets.

**Definition 4 (*Parsing Length* $N_{\mathbb{B}}$)**

*A parsing length $N_{\mathbb{B}}$ of a sequence parsed by a belief set $\mathbb{B}$ is the length of the parsing result after the sequence has been parsed by chunks within $\mathbb{B}$.*

**Definition 5 (*Count Function* $N_{\mathbb{B}}(c)$)**

*$N_{\mathbb{B}}(c)$ denotes the count of how many times the chunk $c$ in the belief set $\mathbb{B}$ appears in the parsed sequence.*

The parsing length and the count function for all of the chunks $c$ on a belief set $\mathbb{B}$ satisfy the following relation:

$$N_{\mathbb{B}} = \sum_{c \in \mathbb{B}} N_{\mathbb{B}}(c) \tag{3}$$

**Definition 6 ($N_{\mathbb{B}}(x \rightarrow y)$)**

*The number of times chunk $x$ is being parsed following the parse of chunk $y$. $x$ and $y$ are both chunks in the belief set $\mathbb{B}$.*

For any chunk $x$ within any belief set $\mathbb{B}$, one can relate $N_{\mathbb{B}}(x)$ with $N_{\mathbb{B}}(x \rightarrow y)$ by:

$$N_{\mathbb{B}}(x) = \sum_{y \in \mathbb{B}} N_{\mathbb{B}}(x \rightarrow y) \tag{4}$$

When the length of the sequence becomes infinite, it is easier to work with probabilities instead of the count function. One can define a probability space over the belief set:

**Definition 7 (*Probability space of a belief set*)**

*With a belief set $\mathbb{B}$, one can define a associated probability space $(\mathcal{S}_{\mathbb{B}}, \mathcal{F}_{\mathbb{B}}, \mathcal{P}_{\mathbb{B}})$. $\mathcal{S}_{\mathbb{B}}$ is the sample space representing all of the possible outcomes of a chunk parse.*
*An event space $\mathcal{F}$ is the space for all possible sets of events. $\mathcal{F}$ contains all the subsets of $\mathcal{S}_{\mathbb{B}}$.*
*Additionally, the probability function $P_{A_{\mathbb{B}}} : \mathcal{F}_{\mathbb{B}} \rightarrow \mathbb{R}$ is defined on the event space $\mathcal{S}_{\mathbb{B}}$. The probability function $P_{A_{\mathbb{B}}}$ satisfies the basic axioms of probability:*

- *$P_{A_{\mathbb{B}}}(E) \geq 0 \ \forall E \in \mathcal{F}$. For any subset in the event space, the probability of an observation being in the subset is positive.*

- *$M, N \in \mathcal{F}$, and $M \cap N = \mathbb{E}$, then $P(M \cup N) = P(M) + P(N)$. For two non-intersecting subsets in the event space, the probability of observing any element that falls within the union of the two subsets is the sum of the probability of observing any event within one subset and the probability of observing any event from the other subset.*

- *$P(S) = 1$. The probability of observing any event that belongs to the sample space is one.*

The probability of a chunk $c$ on a support set of chunks $\mathbb{B}$, is the limiting case of this ratio when $N_{\mathbb{B}}$ goes to infinity.

$$P_{\mathbb{B}}(c) = \lim_{N_{\mathbb{B}} \to \infty} \frac{N_{\mathbb{B}}(c)}{N_{\mathbb{B}}} \tag{5}$$

A learning model keeps track of the occurrence probability associated with each chunk in the belief set. For a current belief set, the model assumes that the chunks within the belief set occurs independently.

The occurrence probability of chunk $c_i$ with the belief set $\mathbb{B}_d$ is $P_{\mathbb{B}_d}(c_i)$, which refers to the normalized frequency of observing chunk $c_i$ when the number of parsing the sequence using chunks from the belief set goes to infinity. In this way, the probability of observing a sequence of chunks $c_1, c_2, ....c_N$ can be denoted as $P(c_1, c_2, ....c_N)$. The joint probability of observing any chunk in the generative process is:

$$P(c_1, c_2, ....c_N) = \prod_{c_i \in \mathbb{B}_d} P_{\mathbb{B}_d}(c_i) \tag{6}$$

In this formulation, chunks as observation units serve as independent factors that disentangle observations in the sequence.

**Definition 8 (*Marginal Parsing Frequency $M_d$*)**

*A vector that stores the number of parses for each chunk $c$ in the belief set $\mathbb{B}_d$.*

$M_d$ contains size $|\mathbb{B}_d|$. Additionally the model keeps track of the transition probability from one chunk to another, as they are used to test whether two chunks have immediate temporal adjacency association.

**Definition 9 (*$T_d$*)**

*The set of transition frequency from any chunk $c_i \in \mathbb{B}_d$ to $c_j \in \mathbb{B}_d$*

**Definition 10 (*Chunk Hierarchy Graph $\mathcal{G}_d$*)**

*The relation between chunks and their constructive components in the generative model is described by a chunk hierarchy graph $\mathcal{G}_d$ with vertex set $V_{\mathbb{A}_d}$ and edge set $E_{\mathbb{A}_d}$. One example of a chunk hierarchy graph is illustrated in Figure 1 a). In this hierarchical generative model, $d$ is the depth of the graph and $\mathbb{A}_d$ is the set of chunks used as atomic units to construct the sequence. Each vertex in $V_{\mathbb{A}_d}$ is a chunk, and edges connect the parent chunk vertices to their child chunk vertices.*

As the belief set $\mathbb{B}$ keeps changing when one modifies the chunks in a sequence, so does the parsing length $N_{\mathbb{B}}$ and the probability associated with the belief set $P_{A_{\mathbb{B}}}$. **Based on the definition of N on how chunks are parsed when the support set changes, this change of N changes a set of constraints on the probabilities defined on the new, augmented support set. To do this, we need to:**

- Formulate the definition of probability based on N.

- Identify all relevant changes of N before and after the chunk update.

- Translate this change of N to the constraints on probability updates.

We derive the relation between the probabilities when two chunks $c_L$ and $c_R \in A_d$ are concatenated together to form a new chunk $c_L \oplus c_R$ and update the alphabet to $A_{d+1}$.

### D.2    IDENTIFY N UPDATES

This is how the count function changes before and after the update. We start with the update of the summary N when the alphabet goes from $\mathbb{A}_d$ to $\mathbb{A}_{d+1}$, and proceed to the update for the marginal N, and then the transitional N.

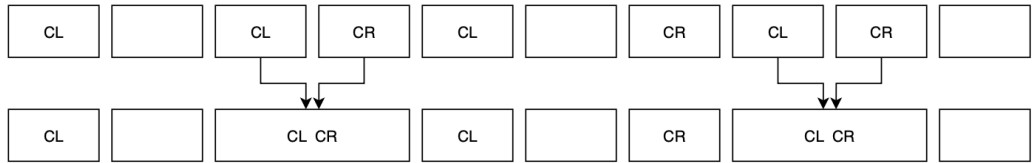

### D.2.1 SUMMARY N

When going from $\mathbb{A}_d$ to $\mathbb{A}_{d+1}$, $c_L$ and $c_R$ are both chunks in $\mathbb{A}_d$ and merged together as a new chunk to augment $\mathbb{A}_d$.

The number of parses changes only in places where the $c_L$ and $c_R$ associated with the new chunk occurs. The chunks in $\mathbb{A}_d$ can be divided into three groups, $c_L$, $c_R$, and $\mathbb{A}_d \setminus \{c_L, c_R\}$. The count function associated with chunks from the set $\mathbb{A}_d \setminus \{c_L, c_R\}$ does not change when $\mathbb{A}_d$ updates to $\mathbb{A}_{d+1}$.

Since with the chunk update $\mathbb{A}_{d+1}$, $c_L$ and $c_R$ are chunked together, they are recognized together as a whole, so every time when they are recognized together as a whole, the count reduces twofold. $N_{d+1}(c_L)$ and $N_{d+1}(c_R)$ are the number of counts for $c_L$ and $c_R$ when they do not occur together. This count is different from $N_d(c_L)$ and $N_d(c_R)$ because the occasions when they occur one after another is taken into the count by $N_{d+1}(c_L \oplus c_R)$. The relation between $N_d$ and $N_{d+1}$ is:

$$N_{d+1} = \left[ \sum_{c \in \mathbb{A}d - c_L - c_R} N_d(c) \right] + N_{d+1}(c_L) + N_{d+1}(c_R) + N_{d+1}(c_L \oplus c_R) \tag{7}$$

$N_{d+1}(c_L)$ could occur in the case where $c_R$ does not follow $c_L$.

$$N_{d+1}(c_L \oplus c_R) = N_d(c_L \rightarrow c_R) \tag{8}$$
$$N_{d+1}(c_L) = N_d(c_L) - N_d(c_L \rightarrow c_R) \tag{9}$$
$$N_{d+1}(c_R) = N_d(c_R) - N_d(c_L \rightarrow c_R) \tag{10}$$
$$N_{d+1}(c_L \oplus c_R) * 2 + N_{d+1}(c_L) + N_{d+1}(c_R) = N_d(c_L) + N_d(c_R) \tag{11}$$

Chunking reduces the number of times sub-chunks are being parsed when sub-chunks occur right after each other by twofold.

$$N_d = \sum_{c \in \mathbb{A}_d} N_d(c) = \left[ \sum_{c \in \mathbb{A}_d - c_L - c_R} N_d(c) \right] + N_d(c_L) + N_d(c_R) \tag{12}$$

Comparing the above two equations we arrive at

$$N_d - N_d(c_L) - N_d(c_R) = N_{d+1} - N_{d+1}(c_L) - N_{d+1}(c_R) - N_{d+1}(c_L \oplus c_R) \tag{13}$$

Since:

$$N_d(c_L) + N_d(c_R) = N_{d+1}(c_L) + N_{d+1}(c_R) + 2N_{d+1}(c_L \oplus c_R) \tag{14}$$

This is related to:

$$N_d(c_L \rightarrow c_R) = N_{d+1}(c_L \oplus c_R) \tag{15}$$

Because both are counting the number of times they occur consecutively.

$$N_{d+1}(c_L) + N_{d+1}(c_R) = N_d(c_L) + N_d(c_R) - 2N_d(c_L \rightarrow c_R) \tag{16}$$

One can define $N_{d+1}$ in terms of counts in $N_d$ as:

$$N_{d+1} = \left[ \sum_{c \in \mathbb{A}_d - c_L - c_R} N_d(c) \right] + N_d(c_L) + N_d(c_R) - 2N_d(c_L \rightarrow c_R) + N_d(c_L \rightarrow c_R)$$

$$N_{d+1} = \left[ \sum_{c \in \mathbb{A}_d - c_L - c_R} N_d(c) \right] + N_d(c_L) + N_d(c_R) - N_d(c_L \rightarrow c_R) \tag{17}$$

Generally, the result is also the case with the total count $N_d$ and $N_{d+1}$ when one switches from the alphabet set $\mathbb{A}_d$ to $\mathbb{A}_{d+1}$ by chunking $c_L$ and $c_R$ in $\mathbb{A}_d$ together.

$$N_{d+1} = \left[ \sum_{c \in A_d - c_L - c_R} N_d(c) \right] + N_d(c_L) + N_d(c_R) - N_d(c_L \to c_R) \tag{18}$$

$$N_{d+1} = N_d - N_d(c_L \to c_R) \tag{19}$$

### D.2.2 Marginal N

To proceed into formulating the joint and conditional probability given a particular belief space, we need to formulate how the count of $N(c)$ for a chunk changes when the belief space when switching from $A_d$ to $A_{d+1}$, with the same division as before.

Of course, the count function should be fixed. However, the probability function associated with the chunks will change based on the update of the belief set. We use the update of the count function to find the relation between the probability updates.

For all $x$ in $\mathbb{A}_d - \{c_L, c_R, c_L \oplus c_R\}$:

$$N_{d+1}(x) = N_d(x) \tag{20}$$
$$N_{d+1}(c_R) = N_d(c_R) - N_d(c_L \to c_R) \tag{21}$$
$$N_{d+1}(c_L) = N_d(c_L) - N_d(c_L \to c_R) \tag{22}$$
$$N_{d+1}(c_L \oplus c_R) = N_d(c_L \to c_R) \tag{23}$$

### D.2.3 Transitional N

The following relationship needs to hold:

for all $x, y$ in $\mathbb{A}_d \setminus \{c_L, c_R, c_L \oplus c_R\}$:

$$N_{d+1}(x \to y) = N_d(x \to y) \tag{24}$$
$$N_{d+1}(x \to c_L) = N_d(x \to c_L) - N_{d+1}(x \to c_L \oplus c_R) \tag{25}$$
$$N_{d+1}(x \to c_R) = N_d(x \to c_R) \tag{26}$$
$$N_{d+1}(x \to c_L \oplus c_R) = N_{d+1}(x) - N_{d+1}(x \to c_R) - N_{d+1}(x \to c_L) - N_{d+1}(x \to y) \tag{27}$$
$$N_{d+1}(x \to c_L \oplus c_R) \leq N_d(c_L \to c_R) \tag{28}$$

From $c_L$ we have this relation.

$$N_{d+1}(c_L \to x) = N_d(c_L \to x) \tag{29}$$
$$N_{d+1}(c_L \to c_L) = N_d(c_L \to c_L) - N_{d+1}(c_L \to c_L \oplus c_R) \tag{30}$$
$$N_{d+1}(c_L \to c_R) = 0 \tag{31}$$
$$N_{d+1}(c_L \to c_L \oplus c_R) = N_{d+1}(c_L) - N_{d+1}(c_L \to c_R) - N_{d+1}(c_L \to c_L) - N_{d+1}(c_L \to x) \tag{32}$$
$$N_{d+1}(c_L \to c_L \oplus c_R) \leq N_d(c_L \to c_R) \tag{33}$$
$$N_{d+1}(c_L \to c_L \oplus c_R) \leq N_d(c_L \to c_R) \tag{34}$$

From $c_R$ we have the following relation:

$$N_{d+1}(c_R \to y) = N_d(c_R \to y) - N_{d+1}(c_L \oplus c_R \to y) \tag{35}$$
$$N_{d+1}(c_R \to c_L) = N_d(c_R \to c_L) - N_{d+1}(c_L \oplus c_R \to c_L) \tag{36}$$
$$N_{d+1}(c_R \to c_R) = N_d(c_R \to c_R) - N_{d+1}(c_L \oplus c_R \to c_R) \tag{37}$$
$$N_{d+1}(c_R \to c_L \oplus c_R) = N_{d+1}(c_R) - N_{d+1}(c_R \to c_R) - N_{d+1}(c_R \to c_L) - N_{d+1}(c_R \to y) \tag{38}$$
$$N_{d+1}(c_R \to c_L \oplus c_R) \leq N_d(c_L \oplus c_R) \tag{39}$$

From the chunked unit $c_L \oplus c_R$ we have the following relation:

$$N_{d+1}(c_L \oplus c_R \to x) \le min(N_d(c_L \oplus c_R), N_{d+1}(c_R \oplus x)) \tag{40}$$

$$N_{d+1}(c_L \oplus c_R \to c_L) \le min(N_d(c_L \oplus c_R), N_d(c_R \oplus c_L)) \tag{41}$$

$$N_{d+1}(c_L \oplus c_R \to c_R) \le min(N_{d+1}(c_L \oplus c_R), N_{d+1}(c_R \oplus c_R)) \tag{42}$$

$$N_{d+1}(c_L \oplus c_R \to c_L \oplus c_R) \le min(N_{d+1}(c_L \oplus c_R), N_d(c_R \oplus c_L)) \tag{43}$$

Finally, we need to satisfy the marginal constraint, that is:

$$N_{d+1}(c_L \oplus c_R) = N_{d+1}(c_L \oplus c_R \to c_L) + N_{d+1}(c_L \oplus c_R \to x) + N_{d+1}(c_L \oplus c_R \to c_R) + N_{d+1}(c_L \oplus c_R \to c_L \oplus c_R) \tag{44}$$

## D.3    PROBABILITY DENSITY SWITCH WHEN $\mathbb{A}_d$ EXPANDS TO $\mathbb{A}_{d+1}$

The constraint is: the number of counts $N$ for all chunks defined for the support set $\mathbb{A}_d$ must remain the same for the support set $sA_{d+1}$, so that the definition of $P_{\mathbb{A}_d}$ for all relevant chunks within $\mathbb{A}_d$ remains the same when $\mathbb{A}_d$ expands to $\mathbb{A}_{d+1}$.

Relating the number of counts to probability: The probability of a chunk occurring in the alphabet set $\mathbb{A}_d$ is defined as:

$$P_{\mathbb{A}_d}(c) = \lim_{N_d \to \infty} \frac{N_d(c)}{N_d} \tag{45}$$

Because $N_d$ and $N_{d+1}$ are only a constant away, both go to infinity if one of them does, so there is a relation between the definition of probability $P_{\mathbb{A}_d}(c)$ and $P_{\mathbb{A}_{d+1}}(c)$. For any chunk $x$ in $\mathbb{A}_d$ that is not $c_L$ and $c_R$, $N_{d+1}(x) = N_d(x)$:

$$P_{A_{d+1}}(x) = \lim_{N_{d+1} \to \infty} \frac{N_{d+1}(x)}{N_{d+1}} = \lim_{N_d \to \infty} \frac{N_d(x)}{N_d - N_d(c_L \to c_R)} \tag{46}$$

That is, the probability of a chunk of this category at d and d+1 satisfies this relationship:

$$\lim_{N_{d+1} \to \infty} P_{A_{d+1}}(x) N_{d+1} = \lim_{N_d \to \infty} P_{A_d}(x) N_d \tag{47}$$

$$P_{A_{d+1}}(x) = P_{A_d}(x) \frac{\lim_{N_d \to \infty} N_d}{\lim_{N_{d+1} \to \infty} N_{d+1}} \tag{48}$$

$$P_{A_{d+1}}(x) = P_{A_d}(x) \frac{\lim_{N_d \to \infty} N_d}{\lim_{N_{d+1} \to \infty} N_d - N_d(c_L \to c_R)} \tag{49}$$

For $c_L$ and $c_R$ in $A_{d+1}$:

$$P_{A_{d+1}}(c_L) = \lim_{N_{d+1} \to \infty} \frac{N_{d+1}(c_L)}{N_{d+1}} \tag{50}$$

$$P_{A_d}(c_L) = \lim_{N_{d+1} \to \infty} \frac{N_d(c_L)}{N_d} \tag{51}$$

$$P_{A_{d+1}}(c_R) = \lim_{N_{d+1} \to \infty} \frac{N_{d+1}(c_R)}{N_{d+1}} \tag{52}$$

$$P_{A_d}(c_R) = \lim_{N_d \to \infty} \frac{N_d(c_R)}{N_d} \tag{53}$$

Because:

$$N_{d+1}(c_L) = N_d(c_L) - N_d(c_L \oplus c_R) \tag{54}$$

$$P_{A_{d+1}}(c_L) = \lim_{N_{d+1} \to \infty} \frac{N_d(c_L) - N_d(c_L \oplus c_R)}{N_{d+1}} \tag{55}$$

$$P_{A_{d+1}}(c_L) = \lim_{N_d \to \infty} \frac{N_d(c_L) - N_d(c_L \to c_R)}{N_d - N_d(c_L \to c_R)} \tag{56}$$

$$P_{A_{d+1}}(c_R) = \lim_{N_d \to \infty} \frac{N_d(c_R) - N_d(c_L \to c_R)}{N_d - N_d(c_L \to c_R)} \tag{57}$$

Finally:

$$P_{A_{d+1}}(c_L \oplus c_R) = \lim_{N_{d+1} \to \infty} \frac{N_{d+1}(c_L \oplus c_R)}{N_{d+1}} \tag{58}$$

$$P_{A_d}(c_L \oplus c_R) = \lim_{N_d \to \infty} \frac{N_d(c_L \to c_R)}{N_d} \tag{59}$$

since $N_d(c_L \to c_R) = N_{d+1}(c_L \oplus c_R)$, we have

$$P_{A_{d+1}}(c_L \oplus c_R) = \lim_{N_d \to \infty} \frac{P_{A_d}(c_L \oplus c_R) N_d}{N_{d+1}} \tag{60}$$

### D.3.1 SUMMARY PROBABILITIES

$$N_{d+1} = N_d - N_d(c_L \oplus c_R) = N_d - N_d P_d(c_L \oplus c_R) \tag{61}$$

$$\frac{N_{d+1}}{N_d} = 1 - P_d(c_L \oplus c_R) \tag{62}$$

### D.3.2 MARGINAL PROBABILITIES

The constraints on marginal probabilities when the support set changes from $A_d$ to $A_{d+1}$, derived from the constraints on the marginal counts, are the following:

$$P_{d+1}(x) = \frac{P_d(x)}{1 - P_d(c_L \oplus c_R)} \tag{63}$$

$$P_{d+1}(c_R) = \frac{P_d(c_R) - P_d(c_L \oplus c_R)}{1 - P_d(c_L \oplus c_R)} \tag{64}$$

$$P_{d+1}(c_L) = \frac{P_d(c_L) - P_d(c_L \oplus c_R)}{1 - P_d(c_L \oplus c_R)} \tag{65}$$

$$P_{d+1}(c_L \oplus c_R) = \frac{P_d(c_L \oplus c_R)}{1 - P_d(c_L \oplus c_R)} \tag{66}$$

With this set of formulations, we have defined the next level marginal probability measures as a function of the previous level observations and their implied probability measures.

### D.3.3 TRANSITIONAL PROBABILITIES

for all $x, y$ in $A_d - \{c_L, c_R, c_L \oplus c_R\}$, from x we have this relation:

$$0 \le P_{d+1}(\cdot|x) \le 1 \tag{67}$$

$$P_{d+1}(y|x) = P_d(y|x) \tag{68}$$

$$P_{d+1}(c_R|x) = P_d(c_R|x) \tag{69}$$

$$P_{d+1}(c_L|x) + P_{d+1}(c_L \oplus c_R|x) = P_d(c_L|x) \tag{70}$$

This equation is sampled so that both $P_{d+1}(c_L|x)$ and $P_{d+1}(c_L \oplus c_R|x)$ satisfy the following constraints:

$$P_{d+1}(c_L|x) \le \frac{P_{d+1}(c_L)}{P_{d+1}(x)} \tag{71}$$

$$P_{d+1}(c_L \oplus c_R|x) \le \frac{P_{d+1}(c_L \oplus c_R)}{P_{d+1}(x)} \tag{72}$$

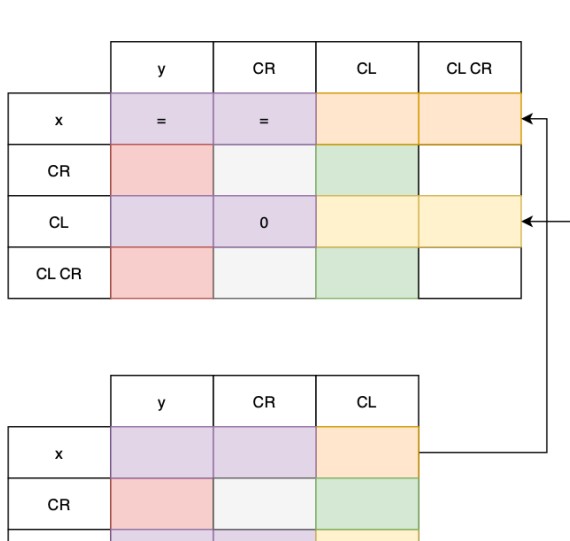

Basically, $P_{d+1}(c_L|x)$ is sampled from the range $\left[0, \min\{1, \frac{P_{d+1}(c_L)}{P_{d+1}(x)}, P_d(c_L|x)\}\right]$.

Additionally, $P_{d+1}(c_L \oplus c_R|x)$ is constrained to be within this range: $\left[0, \min\{1, \frac{P_{d+1}(c_L \oplus c_R)}{P_{d+1}(x)}, P_d(c_L|x)\}\right]$.

From $c_L$ we have this relation:

$$P_{d+1}(x|c_L) = \frac{P_d(x|c_L)}{1 - P_d(c_R|c_L)} \tag{73}$$

$$P_{d+1}(c_R|c_L) = 0 \tag{74}$$

$$P_{d+1}(c_L|c_L) = \frac{P_d(c_L|c_L) - (1 - P_d(c_R|c_L))P_{d+1}(c_L \oplus c_R|c_L)}{1 - P_d(c_R|c_L)} \tag{75}$$

Put into simplified terms:

$$P_{d+1}(c_L|c_L) + P_{d+1}(c_L \oplus c_R|c_L) = \frac{P_d(c_L|c_L)}{1 - P_d(c_R|c_L)} \tag{76}$$

$$P_{d+1}(c_L|c_L) \leq \frac{P_{d+1}(c_L)}{P_{d+1}(c_L)} = 1 \tag{77}$$

$$P_{d+1}(c_L \oplus c_R|c_L) \leq \frac{P_{d+1}(c_L \oplus c_R)}{P_{d+1}(c_L)} \tag{78}$$

Additionally, $P_{d+1}(c_L|c_L)$ is constrained to fall within this range: $\left[0, \min\{1, \frac{P_d(c_L|c_L)}{1 - P_d(c_R|c_L)}\}\right]$.

$P_{d+1}(c_L \oplus c_R|c_L)$ within the range $\left[0, \min\{1, \frac{P_{d+1}(c_L \oplus c_R)}{P_{d+1}(c_L)}, \frac{P_d(c_L|c_L)}{1 - P_d(c_R|c_L)}\}\right]$

From $c_R$ we have the following relation:

$$0 \leq P_{d+1}(\cdot|c_R) \leq 1 \tag{79}$$

$$P_{d+1}(y|c_R) = \frac{P_d(y|c_R)P_d(c_R) - P_{d+1}(y|c_L \oplus c_R)P_d(c_L \oplus c_R)}{P_d(c_R) - P_d(c_L \oplus c_R)} \tag{80}$$

$$P_{d+1}(c_L|c_R) = \frac{P_d(c_L|c_R)P_d(c_R) - P_{d+1}(c_L|c_L \oplus c_R)P_d(c_L \oplus c_R)}{P_d(c_R) - P_d(c_L \oplus c_R)} \tag{81}$$

$$P_{d+1}(c_R|c_R) = \frac{P_d(c_R|c_R)P_d(c_R) - P_{d+1}(c_R|c_L \oplus c_R)P_d(c_L \oplus c_R)}{P_d(c_R) - P_d(c_L \oplus c_R)} \tag{82}$$

$$P_{d+1}(\cdot|c_R) \geq P_{d+1}(\cdot|c_L \oplus c_R) \tag{83}$$

Additionally, $P_{d+1}(c_L \oplus c_R|c_R)$ and $P_{d+1}(c_L \oplus c_R|c_L \oplus c_R)$ needs to satisfy:

$$P_{d+1}(c_L \oplus c_R|c_R) = 1 - P_{d+1}(y|c_R) - P_{d+1}(c_R|c_R) - P_{d+1}(c_L|c_R) \tag{84}$$

$$P_{d+1}(c_L \oplus c_R|c_L \oplus c_R) = 1 - P_{d+1}(y|c_L \oplus c_R) - P_{d+1}(c_R|c_L \oplus c_R) - P_{d+1}(c_L|c_L \oplus c_R) \tag{85}$$

For the generative model, when the alphabet set goes from $\mathbb{A}d$ to $\mathbb{A}d + 1$ by chunking $c_L$ and $c_R$ together, the above constraints associated with chunks in $\mathbb{A}_d$ and $\mathbb{A}_{d+1}$ need to be satisfied.

### D.4 HIERARCHICAL GENERATIVE MODEL

At the beginning of the generative process, the atomic alphabet set $A_0$ is specified. Another parameter, $d$, specifies the number of additional chunks that are created in the process of generating the hierarchical chunks. Starting from the alphabet $A_0$ with initialized elementary chunks $c_i$ from the alphabet, the probability associated with each chunk $c_i$ in $A_0$ needs to satisfy the following criterion:

$$\sum_{c_i \in A_0} P_{A_0}(c_i) = 1 \tag{86}$$

Meanwhile, $P(c_i) \geq 0, \forall c_i \in \mathbb{A}_0$.

We assume that at each step the marginal and transitional probability of the previous steps are known. The next chunk is chosen as the combined chunks with the biggest probability. The order of construction in the generative model follows the rule that the combined chunk with the biggest probability on the support set of pre-existing chunk sets is chosen to be added to the set of chunks.

$$c_L \oplus c_R = \underset{c_L, c_R \in \mathbb{A}_d \backslash \{0\}}{\arg\max} P_{\mathbb{A}_d}(c_L \oplus c_R) \tag{87}$$

Under the constraint that:

$$P_{\mathbb{A}_d}(c_L)P_{\mathbb{A}_d}(c_R) \leq P_{\mathbb{A}_d}(c_L \oplus c_R) \leq \min\{P_{\mathbb{A}_d}(c_L), P_{\mathbb{A}_d}(c_R)\} \tag{88}$$

This can be calculated from the transitional and marginal probability of the previous step.

$$c_L \oplus c_R = \underset{c_L, c_R \in \mathbb{A}_d \backslash \{0\}}{\arg\max} P_{A_d}(c_L \oplus c_R) = \underset{c_L, c_R \in \mathbb{A}_d \backslash \{0\}}{\arg\max} P_{A_d}(c_L)P_{A_d}(c_R|c_L) \tag{89}$$

Once the chunk of the next step: $c_L \oplus c_R$ is chosen, the support set changes from $A_d$ to $A_{d+1}$, which is one size bigger. On the new support set, the marginal probability is specified by the marginal update. That is, the marginal and transitional probability defined on the support set $A_d$ fully specifies the marginal probability of the support set $A_d$. Going from generating a new chunk from the support set $A_d$ to the set $A_{d+1}$, the transitional probabilities between chunks in $A_d$ need to satisfy the following constraints:

- $P_{A_{d+1}}(c_L|x) + P_{A_{d+1}}(c_L \oplus c_R|x) = P_{A_d}(c_L|x)$ for all $x$ in $A_d$
- $P_{A_{d+1}}(c_L|c_L)(1 - P_{A_d}(c_R|c_L)) + P_{A_{d+1}}(c_L \oplus c_R|c_L)(1 - P_{A_d}(c_R|c_L)) = P_{A_d}(c_L|c_L)$
- $P_{A_{d+1}}(y|c_R)(P_{A_d}(c_R) - P_{A_d}(c_L \oplus c_R)) + P_{A_{d+1}}(y|c_L \oplus c_R)P_{A_d}(c_L \oplus c_R) = P_{A_d}(y|c_R)P_{A_d}(c_R)$
- $P_{A_{d+1}}(c_L|c_R)(P_{A_d}(c_R) - P_{A_d}(c_L \oplus c_R)) + P_{A_{d+1}}(c_L|c_L \oplus c_R)P_{A_d}(c_L \oplus c_R) = P_{A_d}(c_L|c_R)P_{A_d}(c_R)$
- $P_{A_{d+1}}(c_R|c_R)(P_{A_d}(c_R) - P_{A_d}(c_L \oplus c_R)) + P_{A_{d+1}}(c_R|c_L \oplus c_R)P_{A_d}(c_L \oplus c_R) = P_{A_d}(c_R|c_R)P_{A_d}(c_R)$

- $P_{d+1}(c_L \oplus c_R | c_R) = 1 - P_{d+1}(y|c_R) - P_{d+1}(c_R|c_R) - P_{d+1}(c_L|c_R)$
- $P_{d+1}(c_L \oplus c_R | c_L \oplus c_R) = 1 - P_{d+1}(y|c_L \oplus c_R) - P_{d+1}(c_R|c_L \oplus c_R) - P_{d+1}(c_L|c_L \oplus c_R)$

In total, there are $|x| + |y| + 3$ degrees of freedom. Since $|x| = |y| = |A_d| - 2$, at each step, there are $2|A_d| - 1$ number of values to be specified.

In practice, after the chunks are specified in $\mathbb{A}_d$, the probability value associated with chunks in $\mathbb{A}_0$ are sampled from a flat Dirichlet distribution, which is then sorted so that the smaller sized chunks contain more of the probability mass and the null-chunk carries the biggest probability mass. Then, the above constraint is checked for the assigned probability on each of the newly generated chunk with their associated alphabet set $\mathbb{A}_i$. This process repeats until the probability drawn satisfies the condition for every newly created chunk.

**Theorem 1** (Marginal Probability Space Conservation). *After the addition of $c_{d,i} \oplus c_{d,j}$ and the change of probability, $P_{A_d}$ is still a valid probability distribution.*

**Proof:**

$$
\begin{aligned}
\sum_{c_{d,k} \in \mathcal{A}_d} P_{\mathcal{A}_d}(c_{d,k}) = & \sum_{c_{d,k} \in \mathcal{A}_{d-1} - c_{d-1,i} - c_{d-1,j}} P_{\mathcal{A}_{d-1}}(c_{d-1,k}) + \\
& + P_{\mathcal{A}_{d-1}}(c_{d-1,i}) - P_{\mathcal{A}_{d-1}}(c_{d-1,j}|c_{d-1,i}) P_{\mathcal{A}_{d-1}}(c_{d-1,i}) \\
& + P_{\mathcal{A}_{d-1}}(c_{d-1,j}) + P_{\mathcal{A}_{d-1}}(c_{d-1,j}|c_{d-1,i}) P_{\mathcal{A}_{d-1}}(c_{d-1,i}) \\
= & \ 1
\end{aligned}
\tag{90}
$$

$\square$

**Theorem 2** (Conditional Probability Space Preservation). *The conditional probability $P_{d+1}(z|c)$ for any $c \in A_{d+1}$ after sampling is still a valid distribution.*

**Proof:** Show by manipulating the equations, that the sum of the conditions on c sums to 1, and each of them is bigger than 0. $\square$

**Theorem 3** (Measure Space Preservation). *Given that at the end of the generative process with depth $d$ one ends up having an alphabet set $\mathbb{A}_d$, the probability space defined on $A_i$, which includes the marginal and joint probability of any chunk and combinations of chunks in $A_i$, $i = 0, 1, 2, \ldots d$, which are predecessor alphabet sets of $A_d$, all values in the set $\mathbb{M}_d$ and $\mathbb{T}_d$ remain the same no matter how the future support set changes according to the generative model.*

**Proof:** By induction.

- Base case: starting from the initialized alphabet set $A_0$, the probability of $P_{A_0}(c), c \in A_0$, and the probability of $P_{A_1}(xy), x, y \in A_0$, for all valid c, x, y, when the alphabet is $A_1$. Going from $A_0$ to $A_1$, $N_0(c)$, $N_0$ and $N_0(x \to y)$ does not change, therefore $P_{A_0}(c)$ and $P_{A_0}(x \to y)$ at the alphabet $A_1$ is the same as that when the alphabet is $A_0$.

- Induction Step: starting from the initialized alphabet set $A_d$, the probability of $P_{A_d}(c), c \in A_d$, and the probability of $P_{A_d}(xy), x, y \in A_d$, for all valid c, x, y, when the alphabet is $A_{d+1}$. Going from $A_d$ to $A_{d+1}$, $N_d(c)$, $N_d$ and $N_d(x \to y)$ does not change, therefore $P_{A_d}(c)$ and $P_{A_d}(x \to y)$ at the alphabet $A_{d+1}$ is the same as that when the alphabet is $A_d$.

$\square$

**Theorem 4.** *The order of $P_{A_i}(xy), x, y \in A_i$ for any $i = 0, 1, 2, \ldots d$ at any previous belief space is preserved throughout the update.*

**Proof:** At the end of the generative process with depth $d$, one ends up having such an alphabet set: $A_d$. The probability space defined on $A_i$, which includes the marginal and joint probability of any chunk and combinations of chunks in $A_i$, $i = 0, 1, 2, \ldots d$ is preserved, hence the order is preserved.

$\square$

The generative process can be described by a graph update path. The specification of the initial set of atomic chunks $\mathbb{A}_0$ corresponds to an initial graph $\mathcal{G}_0$ with the atomic chunks as its vertices. At the i-th iteration, as the generative graph goes from $\mathcal{G}_{\mathbb{A}_i}$ to graph $\mathcal{G}_{\mathbb{A}_{i+1}}$, two none zero chunks $c_L$, $c_R$ chosen from the pre-existent set of chunks $\mathbb{A}_i$ and are concatenated into a new chunk $c_L \oplus c_R$,

augmenting $\mathbb{A}_i$ by one to $\mathbb{A}_{i+1}$. The vertex set also increments from $V_{\mathbb{A}_i}$ to $V_{\mathbb{A}_{i+1}} = V_{\mathbb{A}_i} \cup c_L \oplus c_R$. Moreover, two directed edges connecting the parental chunks to the newly-created chunk are added to the set of edges: $E_{\mathbb{A}_i}$ to $E_{\mathbb{A}_i} = E_{\mathbb{A}_i} \cup (c_L, c_L \oplus c_R) \cup (c_R, c_L \oplus c_R)$. The series of graphs created during the chunk construction process going from $\mathcal{G}_{\mathbb{A}_0}$ to the final graph $\mathcal{G}_{\mathbb{A}_d}$ with d constructed chunks can be denoted as a graph generating path $P(\mathcal{G}_{\mathbb{A}_0}, \mathcal{G}_{\mathbb{A}_0}) = (\mathcal{G}_{\mathbb{A}_0}, \mathcal{G}_{\mathbb{A}_1}, \mathcal{G}_{\mathbb{A}_2}, ..., \mathcal{G}_{\mathbb{A}_d})$.

### D.5  LEARNING THE HIERARCHY

The rational chunking model is initialized with one minimally complete belief set, the learning algorithm ranks the joint probability of every possible new chunk concatenated by its pre-existing belief set, and picks the one with the maximal occurrence joint probability on the basis of the current set of chunks as the next new chunk to enlarge the belief set. With the one-step agglomerated belief set, the learning model parses the sequence again. This process repeats until the chunks in the belief set pass the independence testing criterion.

**Theorem 5** (Learning Guarantees on the Hierarchical Generative Model). *As $N \to \infty$, the chunk construction graph learned by the model $\hat{\mathcal{G}}$ is the same as the chunk construction graph of the generative model: $\hat{\mathcal{G}} = \mathcal{G}$, which entails that they have the same vertex set: $\hat{\mathbf{V}} = \mathbf{V}_{\mathcal{G}}$ and the same edge set: $\hat{\mathbf{E}} = \mathbf{E}_{\mathcal{G}}$. Additionally, the belief set learned by the chunk learning model $\mathbb{B}_d = \mathbb{A}_d$, and the marginal probability evaluated on the learned belief set $M_{\mathbb{B}_d}$ associated with each chunk is the same as the marginal probability imposed by the generative model on the generative belief set $M_{\mathbb{A}_d}$.*

**Proof:**  Given that all of the empirical estimates are the same as the true probabilities defined by the generative model, we prove that starting with $B_0$, the learning algorithm will learn $B_D = A_D$. $A_D$ is the belief set imposed by the generative model. We approach this proof by induction.

**Base Step:** As the chunk learner acquires a minimal set of atomic chunks that can be used to explain the sequence at first, the set of elementary atomic chunks learned by the model is the same as the elementary alphabet imposed by the generative model, i.e. $B_0 = A_0$. Hence, the root of the graph, which contains the nodes without their parents, is the same, $\hat{\mathcal{G}} = \mathcal{G}$; put differently, $\hat{\mathbf{V}}_0 = \mathbf{V}_0$

Additionally, the learning model approximates the probability of a specific atomic chunk as $\hat{P}_{A_0}(a_i)$. As $n \to \infty$, for all chunks $c$ in the set of atomic elementary chunks in $B_0$, the empirical probability evaluated on the support set is the same as the true probability assigned in the generative model with the alphabet set $\mathbb{A}_0$:

$$\hat{P}_{\mathbb{B}_0}(c) = \lim_{n \to \infty} \frac{N_0(c)}{N_0} = P_{\mathbb{A}_0}(c) \tag{91}$$

**Induction hypothesis**: Assume that the learned belief set $\mathbb{B}_d$ at step $d$ contains the same chunks as the alphabet set $\mathbb{A}_d$ in the generative model.

The HCM, by keeping track of the transition probability between any pairs of chunks, calculates $\hat{P}_{\mathbb{B}_d}(c_i | c_j)$ for all $c_i$, $c_j$ in $\mathbb{B}_d$. Afterwards, it finds the pair of chunks $c_i$, $c_j$, such that the chunk created by combining $c_i$ and $c_j$ together contains the maximum joint probability violating the independence test as candidate chunks to be combined together.

$$\hat{P}_{B_d}(c_i \oplus c_j) = \sup_{c_i, c_j \in B_d} \hat{P}_{B_d}(c_i) \hat{P}_{B_d}(c_j | c_i) \tag{92}$$

We know that in the generative step the supremum of the joint probability with the support set $\mathbb{A}_d$ is being picked to form the next chunk in the representation graph, so each step of the process at step $d$ satisfies the condition that:

$$P_{A_d}(c_i \oplus c_j) = \sup_{c_i, c_j \in A_d} P_{A_d}(c_i) P_{A_d}(c_j | c_i) \tag{93}$$

Since $P_{\mathbb{A}d}(c_j | c_i) = \hat{P}_{\mathbb{B}d}(c_j | c_i)$, $P_{\mathbb{A}d}(c_i) = \hat{P}_{\mathbb{B}d}(c_i)$, the chunks $c_i$ and $c_j$ chosen by the learning model will be the same ones as those created in the generative model.

**End step**: The chunk learning process stops once an independence test has been passed, which means that the sequence is better explained by the current set of chunks than any of the other possible next-step chunk combinations. This is the case once the chunk learning algorithm has learned a belief set $\mathbb{B}_d$ that is the same as the generative alphabet set $\mathbb{A}_d$. At this point $\hat{\mathcal{G}} = \mathcal{G}$  □

## E  EXPERIMENT DETAIL: CHUNK RECOVERY AND CONVERGENCE

To test the model's learning behavior on this type of sequential data, random graphs of chunk hierarchies with an associated occurrence probability for each chunk are specified by the hierarchical generative process. To do so, an initial set of specified atomic chunks $\mathbb{A}_0$ and a pre-specified level of depth (new chunks) $d$ is used to initiate the generation of a random hierarchical generative graph $\mathcal{G}$. In the end, the generative model generates a set of chunks $\mathbb{A}$. In total, there are $|\mathbb{A}_0| + d$ number of chunks in the generative alphabet $\mathbb{A}$, with chunk $c$ from the alphabet set having an occurrence probability of $P_{\mathbb{A}}(c)$ on the sample space $\mathbb{A}$.

Once the hierarchical generative model is specified, it is then used to produce training sequences with varying length $N$ to test the chunk recovery.

The rational chunk model is trained on the the sequence $S$ with increasing sizes (from 100 to 3000 with steps of 100) generated by the hierarchical generative graph, and it learns a hierarchical chunking graph $\hat{\mathcal{G}}$. To test how good the representation learned by the chunking graph is compared to the ground truth generative model $\mathcal{G}$, a discrete version of Kullback–Leibler divergence is used to compare the ground truth probability $P_{\mathbb{A}}(c)$ of every chunk on the sample space of the ground truth alphabet, to the learned probability $Q(c)$ of the chunking model.

$$KL(P||Q) = \sum_{c \in \mathbb{A}} P_{\mathbb{A}}(c) \log_2\left(\frac{P_{\mathbb{A}}(c)}{Q_{\mathbb{A}}(c)}\right) \tag{94}$$

While $P_{\mathbb{A}}$ is clearly defined by the generative model, $Q_{\mathbb{A}}(c)$ needs to be calculated from what the model has learned. Note that the set of chunks $\mathbb{B}$ learned by HCM may or may not be the same as $\mathbb{A}$, as $\mathbb{B}$ also augments or shrinks in different learning stages.

To evaluate $Q_{\mathbb{A}}(c)$, the hierarchical chunk graph with their occurrence probability associated with each chunk is used to produce "imagined" sequences of length 1000. Imagined sequences are the sequences that HCM produces based on its learned representations. After that, the occurrence probability of each chunk $c$ in $\mathbb{A}(c)$ is used to evaluate $Q$, comparing the HCM's learned representation with the ground truth.

KL divergence is used to evaluate the deviation of learned representations in the hierarchical chunking model from the original representations on the corresponding support set of the original representations.

For the comparsion, we used the same sequence used for training HCM to train a 3 layer recurrent neural network (one embedding layer with 40 hidden units, one LSTM with drop-out rate = 0.2, and one fully connected layer with batch size = 5, sequence length = 3, epoch = 1, so that the data used for training is the same as N) of the training sequence, and used it to generate imagined sequences which are then used to calculate the KL divergence as before.

To generate the KL divergence plot for Figure 2, sequences with varying size $N$ were used to train HCM. HCM was then used to generate imaginative sequence so that KL measurements can be taken.

## F  EXPERIMENT DETAIL: VISUAL HIERARCHICAL CHUNKS

The visual hierarchical chunks are crafted by hand as binary arrays. The dark pixels correspond to having a array value of 1 and background a value of 0. Each image in the generative hierarchy is 25 dimensional (5 x 5) in the visual domain and size 1 in the temporal domain. An empty array is included to denote no observation. The alphabet $\mathbb{A}$ of the generative model include all 14 the images in the generative hierarchy.

The probability of occurrance for each generative visual chunk is drawn from a flat dirichlet distribution with the empty observation retaining the highest mass, this is to emulate the process that real world observatipons are mostly sparse in the environment.

The parameters $(\alpha_1, .., \alpha_K)$ with $K = |\mathbb{A}|$ are all set to one, and $P_{\mathbb{A}}(c), c_1, ..., c_K \in \mathbb{A}$ are sampled from the probability density function

$$f(x_1, ..., x_k; \alpha_1, ..., \alpha_K) = \frac{1}{\mathbf{B(a)}} \prod_{i=1}^{K} x_i^{\alpha_i - 1}) \tag{95}$$

Where the beta function when expressed using gamma function is: $\mathbf{B(a)} = \frac{\prod_{i=1}^{K} \Gamma(\alpha_i)}{\Gamma(\sum_i^K \alpha_i)}$, and $\mathbf{a} = (\alpha_1, .., \alpha_K)$.

To generate the sequence, images in the hierarchy are sampled from the occurrence distribution and appended to the end of the sequence. As a result, there are visual correlations in the sequence defined by the hierarchy, but temporally, each image slice is i.i.d. from the dirichlet distribution.

Examples of chunk representations learned by the hierarchical chunking model at different stages is shown are collected at the learning stages of t = 10, t = 100, t = 1000 respectively.

## G   EXPERIMENT DETAIL: GIF MOVEMENT

The gif file is converted into an [T x H x W] sized tensor. With $T$ being the temporal dimension of the spatial vision sequence. The entire moment is 10 frames of 25 x 25 images. Each color in the gif file is mapped to a unique integer, with the background having a value of 0. TO construct the training sequence for spatial temporal chunks. In this way, the gif file is converted into a tensor with size 10 x 25 x 25. The entire movement is repeated 100 times and trained on HCM.

## H   EXPERIMENT DETAIL: THE HUNGER GAMES

The first book of *The Hunger Games* is stored as `'test_data.txt'` with the code in the supplementary material. The text file contains approximately 520,000 characters in total.

To convert the book into a temporal sequences, a unique mapping between each character and an integer is created. This sequence of integer is used to train the online HCM, with a forgetting rate of 0.99 and a chunk deletion threshold of $1e^-5$. Empty spaces are also mapped onto a nonzero integer to enable the model to learn chunks with the inclusion of empty spaces. The HCM trains on sequences of 1,000 characters in length at each step. More examples of learned chunks taken from $M$ at different stages of learning are displayed in Table 2. Simple representation examples are taken from chunks learned after 10,000 characters in the book. Intermediate examples are taken after 100,000 characters have been parsed, and Complex chunks are taken when HCM reaches 300,000 characters of the book.

Table 2: Representation Learned in Hunger Games

| Simple Chunks | 'an', 'in ', 'be', 'at', 'me', 'le', 'a ', 'ar', 're', 'and ', 've', 'ing', 'on', 'st', 'se', 'to ', 'i ', 'n ', 'of ', 'he', 'my ', 'te', 'pe', 'ou',' we', 'ad', 'de', 'li', 'oo', 'bu', 'fo', 'ave', 'the', 'ce', 'is', 'as',' il', 'ch', 'al', 'no', 'she', 'ing', 'am', 'ack', 'we', 'raw', 'on the',' day', 'ear', 'oug', 'bea', 'tree', 'sin', 'that', 'log', 'ters', 'wood',' now', 'was', 'even', 'leven', 'ater', 'ever', 'but', 'ith', 'ity', 'if',' the wood', 'bell', 'other' |
|---|---|
| **Intermediate Chunks** | 'old', 'gather', 'as', 'under', 'way.', 'day', 'hunger', 'very', 'death', 'ping', 'the seam', 'add', 'ally', 'king', 'lose', 'sing', 'loser', 'money', 'man who', 'in the', 'says', 'tome', 'might', 'rave', 'even', 'ick', 'wood', 'he want', 'for', 'into', 'leave', 'reg', 'lose to', 'lock', 'where', 'up', 'gale', 'older', 'ask', 'come', 'raw', 'real', 'bed', 'ing for', 'from', 'link', 'few', 'close', 'arrow', 'ull', 'cater', 'this', 'one', 'almo', 'lack', 'shop', 'year', 'ring', 'cause', 'is the', 'ugh', 'eve', 'are', 'the leap', 'lly', 'still', 'heal', 'tow', 'never', 'try', 'prim', 'iting', 'bread', 'ould', ', but', 'Now', 'beg', 'liday', 'arm', 'quick', 'hot', 'men', 'know', 'then', 'bell', 'pan', 'mother', 'only', 'war', 'eak', 'high', 'read', 'district 12', 'can', 'would', 'pas', |
| **Complex Chunks** | 'arent', 'he may', 'I want', 'gather', 'capitol', 'been', 'trip', 'a baker', ', but', 'madge', 'heir', 'mouth', 'you can', 'a few', 'berries', 'fully', 'tribute', 'I can', 'cause of the', 'feel', 'to the baker', 'its not just', 'he want', 'slim', 'hand', 'the ball', 'quick', 'green', 'the last', 'peace', 'off', 'you have to', 'kill', 'of the', 'the back', 'they', 'scomers', 'have been', 'reaping', 'as well', ', but the ', 'cent', 'thing', 'I remember', 'ally', 'though', 'again', 'dont', 'need', 'in the school', 'the pig', 'in our', 'them', 'to remember', 'fair', 'bother', 'at the', 'older', 'the square', 'I know', 'house', 'its not', 'once', 'what was', 'out of', 'it is not just', 'our district', 'too', 'I have', 'it out' |

