# OpenReview forum: "Learning Structure from the Ground up---Hierarchical Representation Learning by Chunking"
_ICLR.cc/2022/Conference — ICLR 2022 Submitted_

### Official Review · Reviewer_LNDm · 2021-10-27

**Correctness:** 3
**Technical Novelty And Significance:** 1
**Empirical Novelty And Significance:** 2
**Recommendation:** 3
**Confidence:** 4

**Main Review:**


The paper is clear. I rarely had trouble following, although I didn’t understand that the decision to chunk was based on a chi^2 test until I read the appendix, which seems crucial. I enjoyed reading about the relative sample efficiency of the classical algorithm vs the RNN, though I would have rather seen a fair comparison with a TreeRNN or some other system that involves latent tree structure, as well as a comparison to other classical dependency parsers. The application of a classic parsing algorithm to video was a nice adaptation.

The overall problem I had with this paper was the fact that it is presenting a classic parsing algorithm but contains no citations to any work from the age of classic parsing algorithms. I found this lack of background disturbing because as far as I can tell this algorithm is a statistical stack based parser, and the authors should have looked into whether they were reproducing existing work. The problems they have with efficiency of their own algorithm are resolved by many statistical parsing algorithms. Even allowing partial parses (as they do) is a property in a number of non neural parses such as https://www.cs.cmu.edu/~nschneid/twparser.pdf (A Dependency Parser for Tweets by Kong et al.). Ironically, I also had trouble looking for specific classical parsing algorithms to compare with this while reviewing, because the literature has exclusively contained neural parsing algorithms for so long. The general area of structured prediction is one that has a long history, and the authors seem not have a particular background in the problem space. I recommend reading Slav Petrov’s thesis (https://www2.eecs.berkeley.edu/Pubs/TechRpts/2009/EECS-2009-116.pdf) for a deep background on the topic from the age of classical parsing.

Although the paper described the problem of a lack of inductive bias towards hierarchical parse structures, there were no citations to the literature which attempts to resolve this problem (treeRNNs, RNNGs, etc.). There was also no discussion of non-neural hierarchical algorithms for structured prediction on video (e.g., structured prediction cascades), which seems necessary in a paper with experiments on a non neural hierarchical algorithm for video.

Beyond the lack of discussion of the existing field of algorithmic hierarchical parsing, the discussion of limitations does confront the possibility of non projective grammars, which cannot be covered by this sort of chunking (“How to relax the adjacency assumption as a grouping criterion to allow for non adjacent relationships to be chunked together remains an open challenge”), but does not to discuss it in terms that have been used historically in parsing, or acknowledge the existing parsers that cover non-projective cases. I was somewhat confused by the decision to use The Hunger Games as a corpus for training natural language parsers on, as there are a number of more common corpora that would have compared more easily to the existing literature (The Little Prince, PTB, or wikitext come to mind). I was confused by the reference to Teh 2006 alone as an extension of ngram models, given that there was no other discussion of backoff (e.g., Katz back-off, or smoothing) in ngram models, which has a much longer history.

I was not surprised that introducing a parsing algorithm with a strong inductive bias was more sample efficient than using an RNN. This phenomenon is the reason why, for years, NLP did not use neural networks until large quantities of data and compute became easily available.

MINOR

Please explain how the hypothesis testing works in the main text of the paper, and not just in the appendix, or at least emphasize appendix A in the main text of the paper while describing the algorithm.

Typos:
“they the way”
Hinton (1979) should be a parenthetical but is instead inline citation.

 Questions:

How does catastrophic interference relate to gradient starvation?

**Summary Of The Paper:**

This paper proposes a non neural system of parsing natural language text by chunking sequences to form hierarchical structures. The algorithm strongly resembles classical parsing algorithms. Decisions about when to chunk a phrase into a constituent are based on chi^2 tests of independence, where a pair of chunks that are considered to be dependent are joined into a single constituent. They test this chunking algorithm on natural language data against an RNN,  concluding that the classical parsing algorithm is more sample efficient in achieving a low KL-divergence from the true sequence data. They also provide some examples of how this algorithm can be applied to temporal image data or video.

**Summary Of The Review:**

This paper is missing significant background on classic hierarchical structured prediction. Because it is presenting a classical parsing algorithm without a single citation to pre-neural structured prediction as a field, I believe that it is extremely similar to existing algorithms that are rarely in use today.

---

> ### Author Response · Authors · 2021-11-21
> **Difference between HCM and Classical Parsing Algorithms**
>
> Thank you for the pointers to this part of the literature. We agree that the graphs showing how chunk representations build up bear resemblance to grammar parsing trees in the parsing literature produced by classical NLP research. However, from reading all of the references that the reviewer pointed out, we conclude that our work is motivationally and algorithmically substantially different from these algorithms.
>
> Parsing algorithms use a grammar to parse sentences, the minimal unit of parsing is words, whereas in HCM, the minimal units are syllables, which include blank spaces, symbols, and punctuations. One of the chunks learned by HCM is patterns of punctuations such as ellipsis “...”. This is due to "." and ".." occurring together often, and therefore they are combined together as a whole. In HCM, the unit of parsing grows bigger as chunk learning proceeds, which is a main part of our algorithm but not a feature for fragment grammars or n-gram models. In essence, HCM learns from the lowest level of letters. In that case, a large enough model would represent both morphology (word-level grammar) and syntax (sentence-level grammar), as opposed to parsing algorithms that are only concerned with syntax and require preprocessed word tokens.
>
> Additionally, our model and the parsing model have different goals. The goal of classical parsing algorithms is to parse sentences into a tree-like dependency structure or constituency structure, conforming to the rules of a formal grammar such as X bar grammar. Here, there are multiple possible parses for a given input that fit into the word properties, and the goal of parsing is to find a tree structure consistent with how a human would parse such a sentence. Usually they demand synthetic annotation for a language corpus, and therefore demand annotated datasets such as PTB, The Little Prince, etc… A precondition for parsing words into grammar trees is that each word should be annotated with its properties. These annotations become the latent structure to be parsed with the grammar parse tree.
>
> The goal of HCM is to learn repeated patterns from sequence data, starting with no representation whatsoever and building up the representation by reusing previously learned representations. Anything that can be organized in a sequence of discrete array structures can be used for training HCM. Thereby HCM does not require datasets with annotated treebanks. It also does not parse sentences based on the latent properties of word structures. This is why we currently cannot see the resemblance between our model and parsing models the reviewer is pointing out. However, we are always happy to receive more pointers to relevant models in the literature.
>
> We believe that it is nonetheless a good idea to integrate the papers linked by the reviewer into a related literature section, also to make sure that such misunderstanding do not occur for future versions of our manuscript.
>
> Regarding ngram models, the reason why Teh (2006) was mentioned only is because it is a principled method combining backoff and smoothing, and it’s been shown to make up for the shortcomings of the classic methods based on heuristic backoff or smoothing alone. For instance, in Katz-backoff, a heuristic choice is made about whether to regard a one smaller ngram. As such, if the parameter values are not optimal, useful information can be disregarded completely. In comparison, Teh’s (2006) model performs the backoff across all ngrams and weighs their predictive information proportionally to their certainty, ensuring data efficiency. It is also more plausible that humans combine predictive information from their working memory window as opposed to deciding what part of the window to consider and what part to ignore. Therefore, both for comparison and the future development of HCM, such principled methods are of most interest.
>
> Additionally, we would not term this algorithm as explicitly “non-neural” but instead as “non-artificial-neural-network”. Currently, HCM includes only little inductive biases other than that chunks that happen close in time can become one. This can be seen as an instantiation of the neural property that neurons that fire together, wire together. In fact, a line of future studies is to construct a neurally plausible HCM. Hence, at the initial stage of this work, we find the explicit term “non-neural” (which was probably used for not using neural networks) a little misleading.
>
> We thank the reviewer for their feedback and pointers to the literature.

---

> > ### Comment · Reviewer_LNDm · 2021-11-22
> > **Unsupervised parsing**
> >
> > I'm sorry that some of my initial references weren't as relevant as I hoped. My field is related, but this kind of work is not trendy (to be clear, I think it's a good thing to bring back unfashionable ideas and resituate them in modern contexts of pervasive deep learning), so I haven't thought about a lot of these papers in a long time. However, while you are correct that most classic parsing literature is focused on annotated parse trees, it's not true that this is the only line of research. I think that you need to dig into the literature on unsupervised parsing. A recent [tutorial](https://aclanthology.org/2021.eacl-tutorials.1/) on the topic is available at EACL but I encourage you to go back further.
> >
> > While I understand that your goal is to develop an improved language model by introducing inductive bias towards latent tree structure, and therefore your goal is not to focus on parsing, much of the unsupervised parsing literature has a similar goal. If you look into it, you may find very closely related previous work, or you may find new ideas to follow through on.

---

> > > ### Author Response · Authors · 2021-11-25
> > > **Thank you for the suggestions!**
> > >
> > > While digging into the tutorial, we found a branch of highly relevant, and beautiful work steaming from ideas dating back to the 70s on unsupervised parsing. This has never came into the horizon of our literature search, partially because many of the work is done in the field of cs, partially because "parsing" as a keyword is a bit distanced from how one would formulate and name this idea, and another reason can be that recent literature are dominated by deep learning approaches. But it is nice to see that this vital question on representation learning has been asked in the distanced past. We are really thankful for your suggestions, this could really enable us to stand on the shoulders of giants.

---

### Official Review · Reviewer_zfT1 · 2021-11-01

**Correctness:** 3
**Technical Novelty And Significance:** 3
**Empirical Novelty And Significance:** 2
**Recommendation:** 5
**Confidence:** 3

**Main Review:**

Strengths:
- This paper is particularly well-written and understandable. I appreciated the intuitive explanations of chunking in cognitive science and its extension to common machine learning use cases like language and visual data. The examples of instances where hierarchical chunk learning could both help and hurt a learned model were well-chosen. The figures effectively demonstrated the training process and the learned representations in each domain. Even the theorems were more interpretable than I typically see, being subdivided and laid out piece by piece.
- The method is reasonably novel and broadly applicable. The paper shows HCM applied to temporal, visual, visuo-temporal, and language domains. Given a domain with some hierarchical structure, a fairly reasonable assumption, this method is able to find that hierarchy with some guarantees. The learned hierarchy itself, as the authors note in the conclusion, could be applied to down-the-line endpoints such as -causal learning.
- This method really leans into explainability/interpretability and could thus be more compatible in human-ML frameworks

Weaknesses:
- While the method is novel and seems to recover structure quite well, the results are not as convincing as I’d like.  To lay this out:
Given a toy generative hierarchical model, HCM is able to more effectively predict sequences than a basic RNN, particularly as the levels of hierarchy increased. Not to be too glib, but I should hope so!

- In an environment where the HCM representations overlap with the underlying model, it outperforms a learned-from-scratch HCM, while in the opposite case, it underperforms. The authors suggest that the nature of the HCM (as compared to something like a DNN) allows users to understand a priori whether their pretrained model will work well, which I agree with
- In toy visual domains with and without temporal correlations, HCM learns reproduces the underlying representations. But how does its ability to reproduce the actual sequences compare with appropriate baselines.

- Finally, HCM is applied to a corpus from the Hunger Games and is able to learn commonly-repeated phrases over time.

My main concern with all of this is the lack of actual baselines. I agree that the models are interpretable and useful, but they aren’t applied to any previously-used datasets or compared (empirically) to other SOTA methods. HCM doesn’t necessarily need to *win* in performance, given its other advantages, but I’d like to see whether it’s competitive

- On a related note, the authors provide both idealized and online HCM algorithms. Even the “online” algorithm, while theoretically tractable, seems practically quite slow, which I assume is why the chosen domains are simple. While the online algorithm seems to work well for these domains, I would imagine the loss of guarantees is more likely to be impactful in harder domains.
It was not clear to me how the chunks were generated until I read the independence tests section in the appendix, and I think that this is too important to push out of the body of the paper. It also introduces the hyperparameter of statistical significance p, which isn’t really discussed.


**Summary Of The Paper:**

This paper proposes a method for learning representations of non- i.i.d. data in terms of hierarchical sets of chunks, inspired by cognitive theories of grouping by proximity. These sets are assembled over time from the initial set of primitive data points by finding correlations between temporally/spatially sequential primitives/chunks and appending to the set. The authors show that this learning method is tractable, has convergence w.r.t. hierarchically-decomposable problems, and learns intuitively and practically reasonable chunk sets.

**Summary Of The Review:**

I like this algorithm and think it has potential. I can see how it can be applied both to standard ML tasks, but also how it could unlock a more symbiotic human-ML collaboration through its interpretability. The motivation and build-up from cognitive science is clear, and all else aside, because of its writing, I felt this paper gave me a lot more valuable insights than most. That said, I’m just not convinced by the current set of experiments. I can’t glean how well HCM will *actually* perform (vs. baselines on standard datasets), particularly the online variant, and I suspect it’s not computationally that practical either. With some of these comparisons added, I think I could accept, but for now it’s a reject from me.

---

> ### Author Response · Authors · 2021-11-21
> **We would like to thank the reviewer for their positive feedback. This made us more optimistic that we might be really onto something with our current model.**
>
>
> > While the method is novel and seems to recover structure quite well, the results are not as convincing as I’d like. To lay this out: Given a toy generative hierarchical model, HCM is able to more effectively predict sequences than a basic RNN, particularly as the levels of hierarchy increased. Not to be too glib, but I should hope so!
>
> Of course, that HCM beats a RNN if the underlying model was the hierarchical generative model can be seen as rather trivial. However, for us, this part was more of a proof of concept showing that HCM converges on the generative model.
>
> > My main concern with all of this is the lack of actual baselines. I agree that the models are interpretable and useful, but they aren’t applied to any previously-used datasets or compared (empirically) to other SOTA methods. HCM doesn’t necessarily need to win in performance, given its other advantages, but I’d like to see whether it’s competitive
>
> This is an excellent point. We are currently trying to come up with appropriate baseline measures to compare our model with SOTA. Let us know if you have suggestions along this line.
>
> We will work on improving the experiments in addition to the computational practicality. Again, thank you very much for your comments.

---

> > ### Comment · Reviewer_zfT1 · 2021-11-25
> > **Follow-up**
> >
> > > Of course, that HCM beats a RNN if the underlying model was the hierarchical generative model can be seen as rather trivial. However, for us, this part was more of a proof of concept showing that HCM converges on the generative model.
> >
> > Yes, with the RNN comparison, I figured it was a proof-of-concept...my overall point was just that a lot of space is spent on proof-of-concept empirical evaluations without enough space spent on full comparisons.
> >
> > > This is an excellent point. We are currently trying to come up with appropriate baseline measures to compare our model with SOTA. Let us know if you have suggestions along this line.
> >
> > Difficult question! I think it's clear that your method is able to reconstruct meaningful parts of hierarchical structure (either groupings of pixels, words, or abstract components). There are three avenues that I'm interested in for comparison:
> > 1) How do other methods for identifying hierarchical structure compare? This would primarily be qualitative. For instance, you could compare your learned representations for some text corpora with those generated by Bayesian N-gram, HMM, or inducing models. For methods that used pre-specified chunks on a given dataset, how do your learned representations compare?
> >
> > 2) How well do your learned representations perform in a downstream task? This is a harder one to come up with good examples for. I think the best showcase for your method would be something like video/text prediction in a data-constrained environment. For instance, for the hunger games example, could we take those learned representations and use them as the possible inputs/outputs of a prediction algorithm (instead of primitive words)? The metric on this one is admittedly tough: is it prediction span, prediction accuracy, or a mix of the two? I definitely can't hold it against the paper if I can't come up with a specific one!
> >
> > I'll say upfront that I don't think this is within the scope of the paper, but as an RL researcher, the aspect of HCM that appeals to me is the potential improvement to learning speed. It's difficult to learn both features (using a CNN on image data) and long-term temporal relations (with an RNN) in RL. HCM could parse raw data into meaningful spatio-temporal chunks (e.g., decomposing sequences of Atari frames) which could be used as input to a simplified RL agent (e.g., MLPs instead of CNNs), reducing the amount of data needed to achieve similar performance.
> >
> > 3) How much time does the algorithm take to actually learn these representations (vs N-gram, deep learning, HMM, etc)? The speed of the algorithm seems to be why it's not applied to some of the standard larger datasets

---

### Official Review · Reviewer_1Nwa · 2021-11-02

**Correctness:** 4
**Technical Novelty And Significance:** 3
**Empirical Novelty And Significance:** 3
**Recommendation:** 6
**Confidence:** 3

**Main Review:**

Strengths:
The paper is very well written, with very clear, intuitive explanations for how their method works, and justifications for the authors’ design choices.

The paper provides several well-considered experiments to demonstrate the HCM method quantitatively and qualitatively. First, purely sequential data is generated from several random (but known) heirarchically-structured graphs and the HCM method is shown to learn this underlying hierarchical structure well, compared to a vanilla RNN. Secondly, the paper verifies that the learned model shows positive and negative transfer to similarly or differently structured heirarchical environments, as might be expected from a chunk learning algorithm. Fianlly, the paper explores how the HCM model performs qualitatively in spatial, spatiotemporal or english-language chunking, with interpretable (although unquantified) results in each.

The connections to animal chunk learning are well thought through. Interestingly, for the case of spatiotemporal chunking, without considering a priori the spatial proximity of pixels, spatially connected chunks are learned. So it is by virtue of the fact that objects tend to move smoothly in space and time that online HCM will learn to group visual spatial chunks smoothly in the height x width plane too. This has really interesting close ties to theories for animal learning of object permanence (although obviously the implementation is very different), as the authors note.

Weaknesses:

The paper mentions that this method should offer more interpretable learned representations, but for what sort of task or application is this envisaged? Regarding the transfer of learned chunks to new data sequences, it seems that a human (or other model) would have to know the underlying generative process of the target data sequence in order to know whether the original learned chunking model should work well in the new setting or not (unless of course, the data is generated from the exact same process as the training data). If a human (or other model) knows that then is it not true that you don’t need the model to do the chunking in the first place?


It would have been nice to see quantitative demonstrations of performance for the spatial, spatiotemporal and language-chunk learning experiments. I appreciate its not immediately obvious what the right metric for this performance would be (at least to me), but if the authors were willing and able to find an appropriate one and use this to compare their method to other chunk-learning algorithms it would definitely strengthen the paper.

In the learning plots vs the vanilla RNN, the paper would also have benefitted from comparisons to other explicit chunk-learning algorithms.


**Summary Of The Paper:**

The paper proposes a graph-learning model (HCM) for learning hierarchical chunks from sequential data. The paper first proposes an idealised HCM method, for which the paper provides learning guarantees via a proof by induction, and an online approximation to this idealised method, which is more computationally feasible and which is used to perform experiments in temporal, visual, visuotemporal and language sequential data domains. The paper demonstrates that the online method learns interpretable chunks at multiple levels of abstraction and demonstrates positive (and negative) transfer to other hierarchically structured environments with similar (and different) structures.

**Summary Of The Review:**

A well-written description of a method for a chunk-learning algorithm, with learning guarantees and qualitative demonstrations of sensible-looking chunks across a variety of domains. Quantification of results was a bit lacking.

---

> ### Author Response · Authors · 2021-11-21
> **Thank you for your feedback.**
>
> Thank you for taking the time to read the paper and give us such constructive feedback. Here is a more in detailed discussion regarding several great points that you have raised.
>
>
> >Weaknesses:
> >The paper mentions that this method should offer more interpretable learned representations, but for what sort of task or application is this envisaged? Regarding the transfer of learned chunks to new data sequences, it seems that a human (or other model) would have to know the underlying generative process of the target data sequence in order to know whether the original learned chunking model should work well in the new setting or not (unless of course, the data is generated from the exact same process as the training data). If a human (or other model) knows that then is it not true that you don’t need the model to do the chunking in the first place?
>
> The underlying assumption of the model is that, if there are repeated patterns in the world, then HCM can be used to extract first, what are these repeated patterns, and secondly, what are the relations between these patterns.
> Regarding the transfer of learned chunks to new sequences, this experiment is meant to be a demonstration of transferable skills across environments. Knowing the generative model of the environment makes this demonstration clearer. However, the point is well-taken that we will need to specify further on what sort of applications this ability could be beneficial. We are still actively thinking about possible domains for this, but one that came to mind was medical imaging, in particular neural data, where the practitioner might have some additional domain knowledge. Please let us know if you have further suggestions.
>
> >It would have been nice to see quantitative demonstrations of performance for the spatial, spatiotemporal and language-chunk learning experiments. I appreciate its not immediately obvious what the right metric for this performance would be (at least to me), but if the authors were willing and able to find an appropriate one and use this to compare their method to other chunk-learning algorithms it would definitely strengthen the paper.
>
> One direction that we are currently thinking about is the prediction span. As that is one difference between HCM and other approaches in the sense that the prediction span increases as chunks get longer and the representations build up. As a result, the scope of predictions also increases. Maybe this can be one way to compare the model with other chunk learning or sequence processing models. Our current hypothesis is that other models could be better in short-horizon predictions, whereas HCM could perform better over longer horizons. However, we will still need to test this hypothesis in future studies. Thank you very much for your feedback.

---

### Official Review · Reviewer_mjpW · 2021-11-04

**Correctness:** 3
**Technical Novelty And Significance:** 2
**Empirical Novelty And Significance:** 2
**Recommendation:** 3
**Confidence:** 4

**Main Review:**

**Strengths**

- I believe the paper's main strength lies in its motivation. I believe the core of the presented idea is compelling, and would be of interest to the community.
- The paper is clearly written, and the method is simple.

**Weaknesses**

- The paper presents primarily qualitative results for the majority of datasets/tasks used. The experiment performed on a text corpora only presents a table of examples with learned chunks, and the visual-temporal experiment only presents a figure with some of the learned visual chunks. It is not clear to me from the presented experiments how to compare this method to alternatives. There is one experiment comparing against an RNN baseline, showing that HCM converges faster — however, RNNs are not the current SOTA in sequence modeling (i.e. why wasn't a transformer model used?).
- I am concerned that the method as currently defined cannot generalize to real world data. HCM parses chunks from a sequence by matching them exactly to subsequences, which to me means that this method groups segments together purely based on form rather than semantics. My perspective is that the promise of hierarchical representations is that you can decompose complex objects and patterns into their parts (e.g. a person into [head, arms, legs] — head into [eyes, ear, nose], etc...). However, in modalities such as vision the same parts can appear with drastically different color values. The paper alludes to this in its Discussion section, but does not present a solution to this problem, which is something that I think would need to be shown.
- Related to my point above, I'm not entirely sure I understood the thesis of the paper in terms of the narrative it is trying to convey, and would appreciate hearing the authors thoughts on this. Is this meant to be received as a paper for the cognitive science community, showing an operationalization of grouping by proximity? Or is it being presented for the machine learning community as a representation learning method for use in downstream tasks? If it's the former, I believe this work would be much more appropriately submitted at a cognitive science conference. If it's the latter, I believe much more empirical evidence of the learned representations' usage needs to be shown.
- The related work section mainly focuses on historical NLP methods, with little discussion over similar methods in computer vision, which I believe is needed given that it presents experiments on visual data. I would suggest works such as:
    - Normalized Cuts and Image Segmentation by Shi and Malik 2000
    - Selective Search for Object Recognition by Uijlings et al. 2012

    as places to start. Additionally, I think work on unsupervised grammar induction could also be relevant here.

**Summary Of The Paper:**

This paper presents HCM, an approach for chunking a sequence of data into a hierarchical representation. More specifically, HCM learns a tree with atomic units (ie the low-level inputs, in this case integers representing things like text characters or quantized pixel values) as the leaves and increasingly complex groupings of them higher up the tree.

HCM learns by iteratively parsing the provided data (ie stream of tokens), in each pass computing marginals for the current set of chunks as well as transition frequencies between them. After updating its marginals and transition frequencies, the two chunks with highest joint probability are combined into one. The process continues until all pairs of chunks pass an independence test.

I believe the main contribution of this paper is in that it presents an idea for interpretable grouping based on the principle of grouping by proximity from cognitive science, and a largely qualitative proof of concept for it.

**Summary Of The Review:**

Although I believe the motivating idea is very compelling, I don't believe this paper is ready for publication. In summary, I believe the paper currently lacks:

- More thorough empirical evaluations comparing against other methods.
- Experiments showing the method's potential for generalizing to more naturalistic data, as well as its usefulness for downstream tasks.
- A more clearly focused narrative motivating why it's appropriate for a venue like ICLR as opposed to a cognitive science publication, as well as more thorough contextualization among related work (particularly comparing against recent alternative methods for this problem).

I thank the authors in advance for their response, and am also interested in seeing other reviewers' thoughts.

---

> ### Author Response · Authors · 2021-11-21
> **Thanks for the suggestions**
>
> Here is a more in detailed discussion regarding several great points that you have raised.
>
> > * The paper presents primarily qualitative results for the majority of datasets/tasks used. The experiment performed on a text corpora only presents a table of examples with learned chunks, and the visual-temporal experiment only presents a figure with some of the learned visual chunks. It is not clear to me from the presented experiments how to compare this method to alternatives. There is one experiment comparing against an RNN baseline, showing that HCM converges faster — however, RNNs are not the current SOTA in sequence modeling (i.e. why wasn't a transformer model used?).
>
> We thank the reviewer for raising this point. It is currently not straightforward to find a good way to compare with SOTA methods, given that the representation learned by HCM is not driven by an accuracy measure or a loss function. The KL divergence from the underlying generative model is a quantitative way of measuring it on the hierarchical generative model. However, for transferring to larger datasets such as language and sequence modeling, it is difficult to obtain a direct measurement of performance for HCM. We are still thinking about how to compare our model with alternative models in a fair way, and are open to further suggestions.
>
> > * I am concerned that the method as currently defined cannot generalize to real world data. HCM parses chunks from a sequence by matching them exactly to subsequences, which to me means that this method groups segments together purely based on form rather than semantics. My perspective is that the promise of hierarchical representations is that you can decompose complex objects and patterns into their parts (e.g. a person into [head, arms, legs] — head into [eyes, ear, nose], etc...). However, in modalities such as vision the same parts can appear with drastically different color values. The paper alludes to this in its Discussion section, but does not present a solution to this problem, which is something that I think would need to be shown.
>
>
> We agree with the reviewer that the current model is not applicable in real-world domains where low-level features of chunks can vary. However, we believe that a probabilistic version of the HCM can be developed. In the current version of the HCM, the primitive elements and chunks vary along one feature, yielding ‘concrete’ chunks such as AB. It is possible to represent each chunk by its ‘anchor feature’ value (e.g. spatial coordinates) and also a distribution over additional feature values (e.g. color). In the image processing scenario that has been brought up by the reviewer, for instance, a head would be represented by a chunk that invariably contains the eyes, ears, nose, etc., but allows for variability in the feature values of those chunks, e.g. color of the eyes can be green, blue, etc. We believe that the HCM is an ideal backbone for a more complex model targeted at real-world problems.
>
> > * Related to my point above, I'm not entirely sure I understood the thesis of the paper in terms of the narrative it is trying to convey and would appreciate hearing the author’s thoughts on this. Is this meant to be received as a paper for the cognitive science community, showing an operationalization of grouping by proximity? Or is it being presented for the machine learning community as a representation learning method for use in downstream tasks? If it's the former, I believe this work would be much more appropriately submitted at a cognitive science conference. If it's the latter, I believe much more empirical evidence of the learned representations' usage needs to be shown.
>
> Currently, our paper is meant as a demonstration of a cognitively-inspired idea to sequential data. The goal of the paper was to propose an alternative way of thinking about data as hierarchical instead of iid, as well as the process of learning as learning from scratch instead of starting out with an assumed structure. Applying our model to visual data will demand more thoughts on how to compute similarities between visual representations. We are still developing this part.
>
> >* The related work section mainly focuses on historical NLP methods, with little discussion over similar methods in computer vision, which I believe is needed given that it presents experiments on visual data. I would suggest works such as:
> >* Normalized Cuts and Image Segmentation by Shi and Malik 2000
> >* Selective Search for Object Recognition by Uijlings et al. 2012
> as places to start. Additionally, I think work on unsupervised grammar induction could also be relevant here.
>
> Thanks a lot for the pointers to the literature. We think that we could use some of these models, in particular normalized cuts, to approach the problem of learning visual representations.

---

> > ### Comment · Reviewer_mjpW · 2021-11-30
> > **Response**
> >
> > Thank you to the authors for a very thoughtful reply, the clarifications are greatly appreciated.
> >
> > I think reviewer zfT1 provided some great suggestions for comparisons with SOTA.
> >
> > With regards to chunking low-level features, perhaps some sort vector quantization method could be of use (e.g. VQ-VAE) for learning a discrete codebook for the chunk atoms.
> >
> > I think that work on comparisons to SOTA/existing methods as well as a solution for chunking low-level features semantically, as the authors have suggested in their reply, would really strengthen the paper. I think at its core, the paper presents a very compelling idea, and I think it would be of interest to the research community if such extensions were to be provided in future versions.

---

### Author Response · Authors · 2021-11-21
**Thank you for your constructive feedback.**


We would like to thank all of the reviewers for taking the time to read our paper, and for providing a lot of constructive feedback and suggestions. We have begun working toward making this project better. However, since many of the proposed changes will demand a substantial amount of work, we have decided to not push for an ICLR article but rather take the time to improve this work.

In particular, we will make the following modifications:
1. We will put the independence testing in the main text as suggested by the reviewers.
2. We will compare our model to further appropriate baseline models, in particular  TreeRNN and transformers.
3. We will run further experiments to demonstrate the usefulness of representation interpretability and transfer and are still happy to receive further recommendations about which experiments to run.
4. We will attempt to apply the model to other domains, for example, neural data.
5. We will work towards speeding up our algorithm to run it on more complex datasets.

Again, we are grateful for the constructive feedback. We have added further comments to each reviewer below and welcome further discussions.

---

### Decision · Program_Chairs · 2022-01-20

**Decision:**

Reject

**Comment:**

This paper develops an approach to learning hierarchical representations from sequential data. The reviewers were very positive about the overall approach, finding it well motivated and interesting with strong potential, and thought that the paper was extremely well written with clear examples throughout. There was a good back-and-forth between the reviewers and the authors, discussing several aspects of the paper and providing constructive suggestions for improvement. In particular, the reviewers suggested improvements in terms of independence testing, comparison to further baselines, further experiments, and other improvements as detailed in the reviews. The authors were extremely receptive of these suggestions, which is to be commended and is very much appreciated, and in a response state that they are planning to take the time needed to revise this paper before publication.